# Nemotron-Flash: Towards Latency-Optimal Hybrid Small Language Models

**Yonggan Fu**[†][*], **Xin Dong**[†][*], **Shizhe Diao**[†], **Matthijs Van Keirsbilck**[†], **Hanrong Ye**[†],
**Wonmin Byeon**[†], **Yashaswi Karnati**[†], **Lucas Liebenwein**[†], **Maksim Khadkevich**[†],
**Alexander Keller**[†], **Jan Kautz**[†], **Yingyan (Celine) Lin**[†‡], **Pavlo Molchanov**[†]
[†]NVIDIA Research, [‡]Georgia Institute of Technology

## Abstract

Efficient deployment of small language models (SLMs) is essential for numerous real-world applications with stringent latency constraints.While previous work on SLM design has primarily focused on reducing the number of parameters to achieve parameter-optimal SLMs, parameter efficiency does not necessarily translate into proportional real-device speed-ups. This work aims to identify the key determinants of SLMs' real-device latency and offer generalizable principles and methodologies for SLM design and training when real-device latency is the primary consideration. Specifically, we identify two central architectural factors: depth–width ratios and operator choices. The former is crucial for small-batch-size latency, while the latter affects both latency and large-batch-size throughput. In light of this, we first study latency-optimal depth–width ratios, with the key finding that although deep–thin models generally achieve better accuracy under the same parameter budget, they may not lie on the accuracy–latency trade-off frontier. Next, we explore emerging efficient attention alternatives to evaluate their potential as candidate building operators. Using the identified promising operators, we construct an evolutionary search framework to automatically discover latency-optimal combinations of these operators within hybrid SLMs, thereby advancing the accuracy–latency frontier. In addition to architectural improvements, we further enhance SLM training using a weight normalization technique that enables more effective weight updates and improves final convergence. This technique can serve as a generalizable component for future SLMs. Combining these methods, we introduce a new family of hybrid SLMs, called Nemotron-Flash, which significantly advances the accuracy–efficiency frontier of state-of-the-art SLMs, e.g., achieving over +5.5% average accuracy, $1.3\times/1.9\times$ lower latency, and $18.7\times/45.6\times$ higher throughput compared to Qwen3-1.7B/0.6B, respectively.

**Models on Hugging Face:** Nemotron-Flash-1B | 3B | 3B-Instruct

## 1 Introduction

Recent advances in large language models (LLMs) have driven remarkable breakthroughs across a wide range of applications and industries. Despite their impressive capabilities, the substantial computational demands and high latency of LLMs present significant barriers to practical deployment, particularly in latency-sensitive scenarios or on resource-constrained hardware. This challenge has intensified the demand for small language models (SLMs).

Most existing SLM designs prioritize parameter reduction to achieve efficiency; however, parameter-efficient models do not necessarily yield proportional latency reductions, especially on hardware AI

---

[*]Co-first author.

39th Conference on Neural Information Processing Systems (NeurIPS 2025).

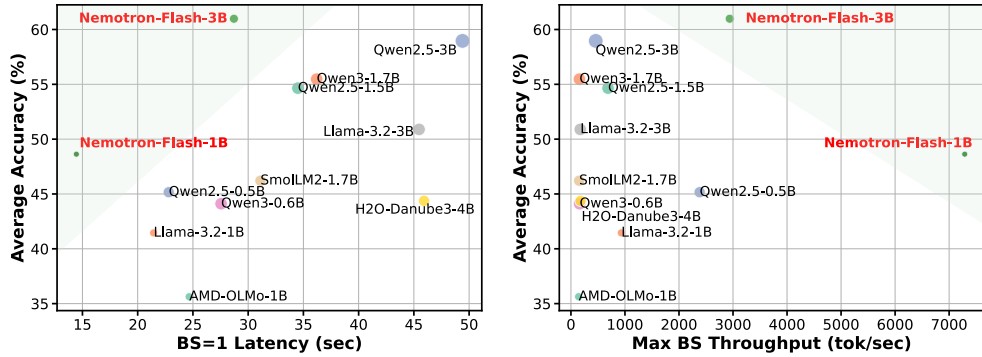

Figure 1: Visualizing (a) the accuracy–latency trade-off and (b) the accuracy-throughput trade-off of our Nemotron-Flash and SOTA SLMs, where the average accuracy is computed across 16 tasks spanning commonsense reasoning, math, coding, and recall tasks. Latency is measured on an NVIDIA H100 GPU for decoding 8k tokens with a batch size of 1 using CUDA Graph. Decoding throughput is measured with a 32k-token input length using the maximum batch size that does not cause out-of-memory (OOM) errors for each model. The marker size represents the model depth.

accelerators like GPUs and TPUs. Additionally, current SLM development processes rely heavily on empirical trial-and-error rather than systematic and principled methodologies. For instance, deep-thin architectures —such as those employed in MobileLLM [1] and SmolLM [2] —often result in suboptimal accuracy-latency trade-offs. Furthermore, with the rapid advent of efficient attention operators [3, 4], the potential synergies of combining these operators in hybrid models have not been thoroughly explored [5, 6, 7, 8], leading to manual, heuristic-driven architecture decisions that are increasingly costly and less scalable.

To address these gaps, this paper introduces generalizable principles and automated methodologies for latency-optimal SLM design and training. We perform a comprehensive study of both architectural choices and training strategies to understand their impact on efficiency and accuracy. Based on these insights, we propose a series of techniques for constructing latency-optimal SLMs and analyze their general applicability and effectiveness. These advancements are integrated into a new family of SLMs, which we refer to as Nemotron-Flash.

Specifically, our architectural exploration focuses on two key factors: depth–width ratios and operator selection, where the former is crucial for small-batch-size latency and the latter affects both latency and large-batch-size throughput. Through extensive training and profiling, we show that (1) deep-thin models, while parameter-efficient, yield suboptimal latency–accuracy trade-offs, and (2) the optimal depth–width ratio scales with the target latency constraints. Guided by this general principle, we also extend existing scaling laws [9] to relate model loss to both depth and width. This allows the sweet-spot depth–width ratio to be determined by profiling a range of configurations and selecting the one that meets the latency constraint while minimizing loss, as predicted by the scaling law.

Further, we comprehensively evaluate emerging attention operators for their accuracy-latency trade-offs and potential as SLM building blocks. With these findings, we introduce an evolutionary search framework that efficiently identifies optimal combinations of hybrid attention operators. Our search strategy exploits the early stabilization of performance rankings among LM architectures, using short training runs as reliable proxies for final performance, thereby enabling fast and reliable search.

In addition to architectural modifications, based on observations of structural patterns in the weight matrices of trained LMs, we further enhance SLM training by constraining weight norms to increase the effective learning rate. This consistently improves final convergence and downstream accuracy across model families. We also employ learnable meta tokens [7] for cache initialization.

By combining these architectural and training innovations, we develop the Nemotron-Flash model family, which significantly advances the accuracy–latency trade-offs for SLMs. As shown in Fig. 1, Nemotron-Flash markedly pushes forward the accuracy–efficiency frontier compared to state-of-the-art (SOTA) SLMs. For example, with all models accelerated using TensorRT-LLM's AutoDeploy kernels [10] and CUDA Graph, Nemotron-Flash-3B achieves +2.0%/+5.5% higher average accuracy, 1.7×/1.3× lower latency, and 6.4×/18.7× higher throughput compared to Qwen2.5-3B/Qwen3-1.7B, respectively. Similarly, Nemotron-Flash-1B achieves +5.5% higher average accuracy, 1.9× lower latency, and 45.6× higher throughput than Qwen3-0.6B.

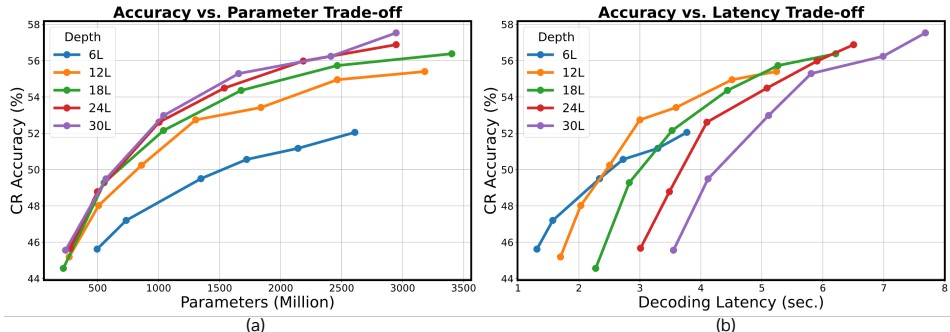

Figure 2: The accuracy–parameter/latency trade-offs when varying depth and width. While deeper models generally achieve a better accuracy–parameter trade-off, they may not perform as well in the accuracy–latency trade-off and there exists an optimal depth-width ratio for a latency budget.

## 2 Towards Latency-Optimal SLM Design and Training

To deliver latency-optimal SLMs, both architecture design and training are critical to achieving the best accuracy-latency trade-off. This work explores both aspects. For SLM design, real-device latency is primarily determined by two key factors: the model's depth and width, and the choice of operators, which are studied in Sec.2.1 and Sec.2.2, respectively. For SLM training, we introduce two optimization techniques in Sec.2.3 and Sec.2.4. Finally, we integrate these components to construct a new model family, Nemotron-Flash, in Sec.3, and benchmark against SOTA SLMs in Sec.3.1.

### 2.1 SLM Design: Depth-Width Ratios

Previous SLMs [1, 11] find that deep-thin models generally achieve better task accuracy than wide-shallow ones under the same parameter budget. However, when targeting real-device latency, this may not hold, as partially noted by [11]. Our key question is: *Do deeper or wider models better optimize the accuracy-latency trade-off?* To answer this, we provide a systematic exploration to understand the impact of depth and width on the accuracy-latency trade-off.

**Exploration settings.** We train a series of Llama models with five depth settings: 6, 12, 18, 24, and 30 blocks, where each block contains one attention and one feed-forward network (FFN), on 100B tokens from the Smollm-corpus [12]. For each depth setting, we also vary the model width (i.e. hidden size) to create models with different sizes and latencies. We visualize the resulting accuracy-parameter and accuracy-latency trade-offs in Fig. 2 (a) and (b), respectively. Accuracy is averaged over eight commonsense reasoning (CR) tasks, and latency is measured as the decoding time for a 1k token generation under a batch size of 1 on an NVIDIA A100 GPU.

**Observations and analysis.** We observe that: ❶ Deeper models generally achieve a better accuracy-parameter trade-off over a wide depth range, although the benefit gradually saturates; ❷ For the accuracy-latency trade-off, the advantage of deep-thin models may not hold, and there exists an optimal depth setting for a given latency budget. For example, when the latency budget is 3 seconds, a depth of 12 achieves the best accuracy among the evaluated settings; ❸ The optimal depth-width ratio generally increases with the latency budget. These observations highlight the necessity of deliberate depth/width selection based on deployment constraints, rather than defaulting to deep-thin models.

**Augmented scaling laws for determining sweet-spot depth-width ratios.** Although the general trend in the above analysis holds, detailed curves may shift across devices and generation lengths, complicating the selection of model depth and width. As such, in addition to the above insights, we also explore principled methods to identify the sweet-spot depth–width ratio within a model family.

In light of this, we augment existing scaling laws [13, 9] by parameterizing model loss with model depth and width. Specifically, existing LM scaling laws [13, 9] parameterize language modeling loss $\mathcal{L}(P, N) = \mathcal{L}_0 + C1 \cdot P^{-\alpha} + C2 \cdot N^{-\gamma}$, with $P$ and $N$ as model size and data size, respectively, and $C1, C2, \alpha, \gamma$ as fitting parameters. We decouple the model size $P$ into two factors, model depth $D$ and width $W$, and reformulate the scaling law:

$$\mathcal{L}(D, W, N) = \mathcal{L}_0 + aD^{-\alpha} + bW^{-\beta} + cN^{-\gamma} \tag{1}$$

where the fitting parameters $a$, $b$, and $c$ control the contributions from each dimension, and the exponents $\alpha$, $\beta$, and $\gamma$ govern the diminishing returns from increasing each respective dimension.

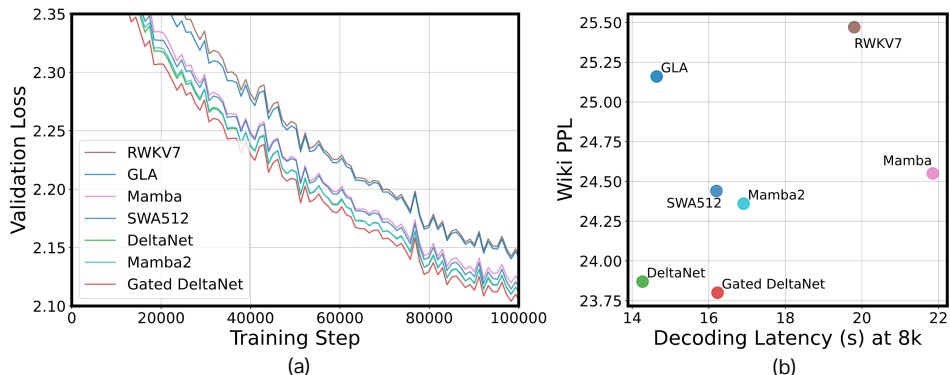

Figure 4: (a) The validation loss evolution, and (b) the PPL-latency trade-off of different operators.

Since the impact of data size is additive and decoupled from depth and width, we can study the effect of depth and width on LM loss *under a fixed data size*, i.e., by neglecting the data size term.

As such, in practice, given a target latency budget and deployment setting, the sweet-spot depth-width ratio can be obtained by profiling a range of depth-width configurations and selecting the one that meets the latency constraint while achieving the lowest loss.

**Fitting and extrapolation.** To validate the effectiveness of this augmented scaling law, we fit it using the above Llama models with varying depth $D$ and width $W$. Specifically, we use perplexity (PPL) as the loss metric, fit the scaling law on a subset of depth/width settings, and validate it on models with larger width/depth settings to assess extrapolation. As shown in Fig. 3, we find that the model extrapolates reasonably well to unseen depth/width settings, staying within 5.3% of the ground-truth PPL, demonstrating that the fitted function generalizes beyond the observed training configurations.

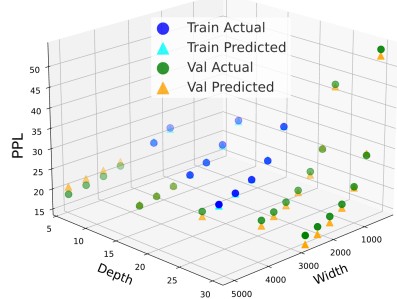

Figure 3: Fitting the scaling law and validating on larger depth/width settings.

**Lesson learned: SLM depth-width ratios.** Deep-thin models may not be latency-optimal, and the optimal depth-width ratio generally increases with the target latency budget. Fitting a scaling law with depth and width provides a more principled approach to identifying the sweet-spot ratios.

## 2.2 SLM Design: Hybrid Operators

Beyond model depth and width, the operator used in each layer is another critical dimension. We first train existing LM architectures in a fully controlled setting to identify the most promising operators on the accuracy-latency frontier. We then develop an evolutionary search pipeline to automatically and efficiently discover hybrid combinations of these operators for constructing hybrid SLMs.

**Exploration settings.** We train a series of 500M LMs built using emerging efficient attention alternatives, including Mamba [14], Mamba2 [15], GLA [16], DeltaNet [17], Gated DeltaNet [18], RWKV7 [19], and sliding window attention (SWA) with a window size of 512. We use the official implementation for Mamba/Mamba2, FlashAttention [20] for SWA, and FlashLinearAttention [21] for all other linear attention variants. All models follow the designs specified in their original papers (e.g., one FFN after each attention operator, except for Mamba/Mamba2), use the same number of blocks, and are trained on 100B tokens from the Smollm-corpus. We present the validation loss in Fig. 4 (a) and the Wikitext PPL-latency trade-off in Fig. 4 (b), measured by decoding 8k tokens with a batch size of 1 on an NVIDIA A100 GPU with CUDA Graph enabled.

In addition, inspired by recent hybrid LMs [6, 22, 5, 23, 7], which combine attention and Mamba/Mamba2 within the same model, we also integrate the promising operators identified in Fig. 4 with Mamba2 or SWA to construct hybrid models in a layer-wise interleaved manner, aiming to gain insights into which operator combinations are well-matched and complementary, as shown in Tab. 1. Note that, for fair comparison, we control the number of blocks in the hybrid models to be the same as in the pure models, based on the insights from Sec. 2.1. More details of the ablation settings are provided in Appendix A.

**Observations and analysis.** We observe that: ❶ In terms of language modeling, DeltaNet and Gated DeltaNet generally emerge as promising candidates, lying on the PPL-latency Pareto frontier; ❷ When integrated into hybrid models with attention or Mamba2, pairing DeltaNet or Gated DeltaNet with Mamba2 typically results in lower PPL and higher accuracy, consistently outperforming the corresponding pure models. In contrast, improvements from pairing with attention are less stable. This indicates both the advantages of hybrid models and the importance of selecting complementary operator combinations; ❸ When used in hybrid models, the performance gap between individual operators may narrow,

Table 1: Exploration of operator combinations. All models have 500M parameters and the same depth.

| Operator1 | Operator2 | ↓ Wiki PPL | ↑ CR Acc (%) |
|---|---|---|---|
| Mamba2 | - | 24.36 | 47.72 |
| Attention | - | 24.44 | 48.02 |
| | *Mamba2* | **23.52** | **48.07** |
| DeltaNet | - | 23.87 | 47.83 |
| | *Attn* | 24.32 | 47.48 |
| | *Mamba2* | **23.37** | **48.03** |
| Gated DeltaNet | - | 23.80 | 47.96 |
| | *Attn* | 24.31 | 47.80 |
| | *Mamba2* | **23.37** | **48.03** |
| GLA | - | 25.16 | 46.82 |
| | *Attention* | 24.60 | 47.63 |
| | *Mamba2* | **24.04** | **47.99** |

likely due to the complementary and diverse memory mechanisms introduced by hybrid layers. For example, although Gated DeltaNet outperforms DeltaNet in language modeling, their task performance becomes comparable when integrated with Mamba2, making DeltaNet the preferable operator in hybrid models due to its greater efficiency.

**Evolutionary search for operator combinations.** The emergence of various efficient attention mechanisms and their complex synergy in hybrid models motivate an automated framework to identify their efficient and complementary combination in hybrid SLMs. To achieve this, we built an evolutionary search engine to efficiently navigate a complex combinatorial design space.

Short-training PPL as a search proxy. A crucial observation underpinning our method is that the relative performance rankings of different LM architectures stabilize early during training, which can also be observed from the validation loss curves in Fig. 4 (a). Using this insight, we demonstrate that *short-training PPL provides a reliable proxy metric to predict final task performance*, substantially reducing training cost for assessing each candidate. We compute the Spearman correlation [24], a measure of rank correlation that is crucial for architecture ranking in our case, between short-training PPL and full-training PPL across multiple LM architectures. We find an 88.8% Spearman correlation, which is sufficient to identify strong architectures within our search space.

Our search space. Based on the identified promising operator candidates and their synergy in hybrid models, we adopt DeltaNet, Attention, and Mamba2 as candidate operators. We search over a maximum of three types of building blocks, each assigned to the early, middle, or late stages of the LM architecture. This three-stage strategy balances operator heterogeneity with architectural regularity. The search explores the ratios of each operator and the number of FFNs per block type, and the repetition count of each block type. More details are provided in Appendix C.1.

Evolutionary search algorithm. We adopt the aging evolution search [25] with the following steps: ① Initialization: Seed and short-train an initial population of architectures, either from known designs or randomly sampled ones; ② Selection: In each evolutionary cycle, we use tournament selection [26] to identify parent architectures that are high-performing based on short-training PPL and meet a predefined target latency budget; ③ Mutation: Selected parents undergo targeted mutations in one of the design factors—operator ratios, FFN ratios, or block type count; ④ Evaluation and

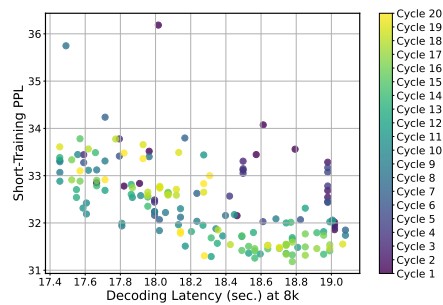

Figure 5: Visualizing the search trajectories.

replacement: Mutated offspring architectures are trained and evaluated using short-training PPL, with latency accurately estimated from a precomputed look-up table (LUT). The oldest architectures are replaced with new candidates, effectively balancing exploration and exploitation. A more detailed description of the algorithm is provided in Appendix C.2.

**Search with decoding latency.** To evaluate the efficacy of our search framework, we conduct a search using decoding latency as the efficiency metric (measured by generating 8k tokens with a batch size of 1 on an NVIDIA A100 GPU). For demonstration purposes, we use a window size of 512

Table 2: Visualizing architectures searched under different efficiency metrics. m2, a, d, and f denote Mamba2, attention, DeltaNet, and FFN, respectively. CR accuracy is averaged over eight tasks.

| Search Metric | Params (M) | ↓ Decoding Latency (8k) | ↓ Wiki PPL | ↑ CR Acc (%) | Searched Operator Combinations |
|---|---|---|---|---|---|
| Decoding Latency | 837 | 17.71 | 20.70 | 51.04 | [d, f, m2, f, a, f, m2, f, d, f, m2, f, a, f, m2, f, d, f, m2, f, d, f, m2, f] |
| Parameters | 497 | 16.94 | 23.06 | 49.23 | [m2, f, a, f, a, f, d, m2, f, a, f, a, f, d, m2, f, a, f, a, f, d, m2, f, a, f, d, f, f] |

for attention, as this length is sufficient for general CR tasks and for the search proxy. We visualize our search process in Fig. 5, where 10 new architectures are sampled and evaluated in each cycle. Our search process progressively improves toward better models with lower PPL under the target latency.

The searched architecture is visualized in Tab. 2. Interestingly, we find that the latency-friendly architecture discovered by the search adopts DeltaNet-FFN-Mamba2-FFN and Attention-FFN-Mamba2-FFN as basic building blocks, stacking them in an interleaved manner. This finding echoes both our earlier observations—that DeltaNet and Mamba2 are strong candidates—and prior work that interleaves attention and state-space models [6, 22, 5, 23, 7].

We benchmark against baseline architectures that have the same model depth and a scaled hidden size to match the decoding latency of our searched architecture. All models are trained on 100B tokens from the Smollm-corpus, and we evaluate them using Wikitext PPL and CR accuracy, averaged over eight tasks. As shown in Tab. 3, the searched hybrid architectures outper-

Table 3: Benchmarking the searched architecture against baselines scaled to the same latency.

| Model | Params (M) | ↓ Wiki PPL | ↑ CR Acc (%) | ↓ Latency |
|---|---|---|---|---|
| SWA | 616 | 23.33 | 48.72 | 18.01 |
| GLA | 862 | 22.67 | 48.43 | 18.19 |
| DeltaNet | 852 | 20.90 | 50.38 | 18.18 |
| Gated DeltaNet | 672 | 21.98 | 49.99 | 17.91 |
| Mamba2 | 601 | 23.14 | 48.61 | 17.82 |
| Mamba2 + FFN | 889 | 21.43 | 50.04 | 17.73 |
| Searched (Ours) | 837 | **20.70** | **51.04** | **17.71** |

form their pure-model counterparts in both PPL and accuracy. This improvement is attributed to: (1) efficient operator combinations that allow for larger parameter counts under the same decoding latency, and (2) the complementary roles played by the hybrid operators.

**Search with number of parameters.** We also conduct a new round of search using the number of parameters (500M) as the efficiency metric. We find that: ❶ The searched architecture achieves over 1.21% higher CR accuracy and a reduction of more than 0.74 in PPL compared to all 500M baselines (detailed results are provided in Appendix C.3); and ❷ as shown in Tab. 2, when comparing the parameter-efficient architecture with the decoding-efficient one, the former generally includes more attention modules, which are parameter-efficient but decoding-unfriendly, and fewer Mamba2/DeltaNet modules, and has greater model depth, which is a parameter-efficient design choice according to Sec. 2.1. This set of experiments validates our search scheme's efficacy in identifying operator combinations that align with the target efficiency metric.

**Lesson learned: SLM operator combinations.** Hybrid models show great promise, but the synergy among different operators is complex, necessitating the identification and combination of complementary operators. The relatively stable ranking of architectures in the early training phases serves as a useful signal for iterating over different designs, and this process can be strategically accelerated using appropriate search algorithms.

## 2.3 SLM Training: Weight Normalization

The potential of SLMs can be better realized when they are properly trained. We observe that model weights trained under a standard scheme exhibit non-smoothness, with large weight magnitudes along certain dimensions, as shown in the first row of Fig. 6. As suggested by [27, 28], larger weight magnitudes can lead to smaller relative weight updates under comparable gradient magnitudes, potentially resulting in ineffective learning during later training stages when the learning rate is low.

Motivated by this and following [28], we constrain weight magnitudes by projecting model weights onto a unit norm sphere after each training iteration. This normalization step eliminates the radial component and emphasizes angular updates, leading to larger relative weight changes under comparable gradient magnitudes. Specifically, based on the patterns shown in Fig. 6, where weight matrices applied to hidden features (noted as *Case-1*) and those whose outputs are added back to hidden features (noted as *Case-2*) exhibit horizontal and vertical patterns, respectively, we perform weight normalization along the corresponding dimensions. Formally, for each weight matrix $\mathbf{W} \in \mathbb{R}^{C_{out} \times C_{in}}$, after each training step, we project it onto a unit norm sphere: $\mathbf{W}_{i,:} \leftarrow \frac{\mathbf{W}_{i,:}}{||\mathbf{W}_{i,:}||_2}$ for $i = 1, \ldots, C_{out}$ for *Case-1*, and $\mathbf{W}_{:,j} \leftarrow \frac{\mathbf{W}_{:,j}}{||\mathbf{W}_{*,j}||_2}$ for $j = 1, \ldots, C_{in}$ for *Case-2*. As shown in the second row of Fig. 6, our weight normalization results in smoother weight distributions.

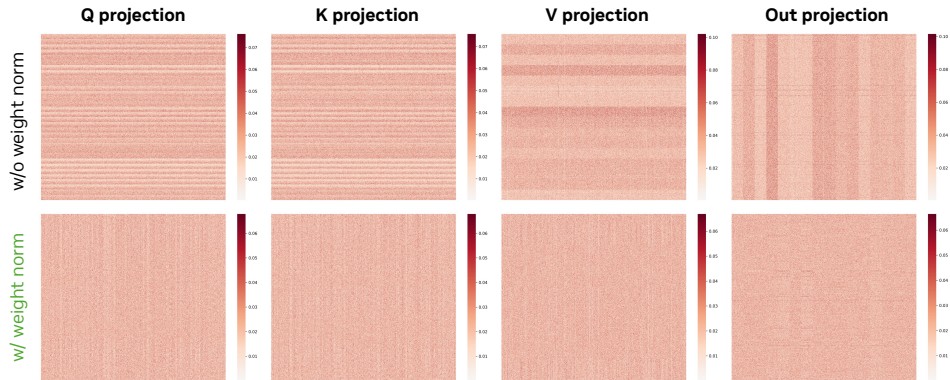

Figure 6: The weight distributions (absolute values) of the layers in the last attention block of 1B Llama models trained without and with our weight normalization technique.

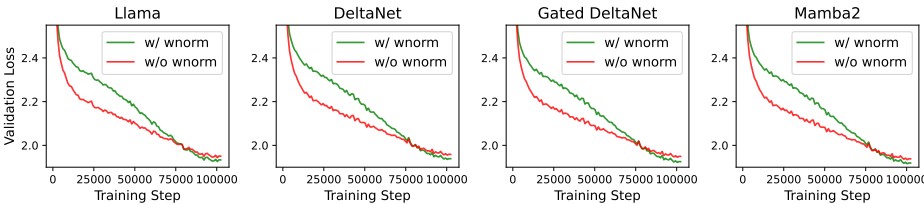

Figure 7: Validation loss trajectories of four model families with and without weight normalization.

**Evaluation across models.** We apply weight normalization to different models and visualize the validation loss curves in Fig. 7, along with the average element-wise gradient norm and L2 norm of the weight matrices in Fig. 8. We can observe that ❶ although in the early training stage, the baseline w/o weight normalization has steeper convergence due to unconstrained weights updates (with radical weight updates), the convergence speed will gradually diminish. With weight normalization, the con-

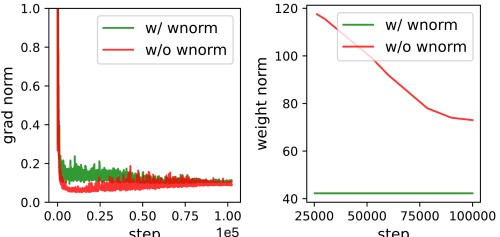

Figure 8: Visualizing the gradient norm and L2 norm of DeltaNet model weights during training.

vergence speed is more constant and the convergence will surpass the baseline in later training stages, leading to consistent better final convergence across model families; ❷ Weight normalization leads to much reduced L2 norm of model weights while slightly increasing the gradient norm compared to the baseline, ensuring larger relative weight changes, which is particularly helpful in late training stages. We observe consistent patterns in other models, provided in Appendix D.2.

We also report the corresponding improvements in language modeling and CR accuracy in Tab. 4. We observe that weight normalization consistently improves CR accuracy by +1.20% and reduces PPL by 0.66 on average across model families, indicating its general effectiveness as an add-on component.

Table 4: Evaluating weight normalization across 1B models trained on 100B tokens from [12].

| Model | Setting | ↓ Wikitext PPL | ↑ CR Acc (%) |
|---|---|---|---|
| Llama 1B | w/o wnorm | 18.67 | 53.81 |
| | w/ wnorm | **18.03** | **54.85** |
| DeltaNet 1B | w/o wnorm | 18.86 | 53.46 |
| | w/ wnorm | **18.19** | **54.39** |
| Mamba2 1B | w/o wnorm | 18.44 | 53.30 |
| | w/ wnorm | **17.88** | **54.71** |

**Connections with nGPT.** Our weight normalization technique can be viewed as a simplified and efficient variant of nGPT [28], which enforces all computations in LMs to operate on a unit sphere by introducing multiple activation normalization layers in addition to weight normalization. We find that: ❶ Weight normalization, along with the resulting more effective weight updates, is key to improved convergence. When applied alone, weight normalization achieves final task performance comparable to the full nGPT solution, as demonstrated in Appendix D.1; ❷ More importantly, the additional activation normalization layers in nGPT introduce significant training overhead—i.e., increasing SLM training time by more than 20%—which reduces the number of training tokens given a fixed training duration. Thus, our contribution lies in identifying the primary contributing component and delivering a more efficient alternative.

## 2.4 SLM Training: Meta Tokens

Previous work [7] has shown that prepending a set of learnable tokens to regular tokens can alleviate attention sinks [29], which are caused by the force-to-attend phenomenon on semantically unimportant tokens. We find that these meta tokens can also benefit non-softmax linear attention mechanisms, as they serve as learned cache initialization when reformulating linear attention in a recurrent format during decoding.

Table 5: Evaluating meta tokens on two linear attention models with 48 operators and on our searched model with 24 operators in Tab. 2.

| Model | Meta Token | ↓ Wikitext PPL | ↑ CR Acc (%) |
|---|---|---|---|
| Mamba2-48L-1B | w/o | 19.04 | 51.71 |
| | w/ | **18.98** | **52.33** |
| DeltaNet-48L-1B | w/o | 19.60 | 52.12 |
| | w/ | **19.47** | **52.46** |
| Searched-24L-830M | w/o | 20.61 | 50.74 |
| | w/ | **20.49** | **51.13** |

As shown in Tab. 5, prepending 256 meta tokens can consistently improve both language modeling and reasoning accuracy (+0.45% on average) with negligible overhead.

## 3 Nemotron-Flash: The New Model Family

Combining all the above architectural improvements and training techniques, we develop and train a new hybrid SLM family called Nemotron-Flash with two different model sizes.

**Model configuration.** We adopt the decoding-friendly model structure searched in Sec. 2.2, which interleaves DeltaNet-FFN-Mamba2-FFN (Block-1) and Attention-FFN-Mamba2-FFN (Block-2) as basic building blocks. We build two models Nemotron-Flash-1B/3B with 0.96B/2.7B parameters, respectively, with depth and width configured based on the scaling laws in Sec. 2.1. Specifically, Nemotron-Flash-1B has the same configuration as in Tab. 2, where the larger parameter count comes from a new tokenizer mentioned below. It has a hidden size of 2048 and contains 12 blocks, each with one token-mixing module and one FFN, or 24 operators if counting DeltaNet, Mamba2, Attention, and FFN as separate operators. Nemotron-Flash-3B has a hidden size of 3072 and contains 36 operators, with two additional Block-1 and one additional Block-2.

**Tokenizer.** Different from previous parameter-efficient SLMs [1] that adopt a tokenizer with a small vocabulary size to save parameters, we adopt a tokenizer with a larger vocabulary size [30]. We find that the latency overhead of the enlarged embedding layer/LM head is small, while the more coarse-grained token representations reduce the token count when encoding the same sentence, resulting in more significant latency reduction. Detailed analysis is in Appendix E.

**Training settings.** Both models are trained using the Adam optimizer (without weight decay, due to the use of weight normalization) and a cosine learning rate schedule with an initial learning rate of 1e-3. We first train the models on Zyda2 [31], then switch to higher-quality datasets, including commonsense reasoning datasets (Climb-Mix [32] and Smollm-corpus [12]), a proprietary high-quality dataset with high proportions of math and code, and MegaMath [33]. Both models are trained for 4.5T tokens using 256 NVIDIA H100 GPUs, with a batch size of 2M tokens and a context length of 4096, except for the final 25B tokens, where we extend the context length to 29000.

**Deployment settings for latency measurement.** To fairly benchmark against baseline models dominated by full attention layers, we adopt TensorRT-LLM's AutoDeploy kernels [10] with efficient KV cache management for full attention and use CUDA Graph for further acceleration. For other operators, we use the official implementation of Mamba2 [15] and FlashLinearAttention [21] for linear attention layers such as DeltaNet, and we always wrap the entire model in a CUDA Graph.

**Baselines and tasks.** We benchmark against SOTA SLMs, including Qwen3 [34], Qwen2.5 [35], Llama3.2 [36], SmolLM2 [2], h2o-Danube [37], and AMD-OLMo [38] series. Accuracy is evaluated using lm-evaluation-harness [39] across 16 tasks including MMLU, commonsense reasoning (PIQA, ARCC, ARCE, Hellaswag, Winogrande, OBQA), math (GSM8k, MathQA), coding (HumanEval, HumanEval-Plus, MBPP, MBPP-Plus), and recall (FDA, SWDE, Squad). We use 5-shot evaluation for GSM8K and MMLU, 3-shot for MBPP and MBPP-Plus, and 0-shot for all remaining tasks. We report the average accuracy on each domain in Tab. 6 and provide task-wise accuracy in Appendix B.

### 3.1 Benchmark with SOTA Base SLMs

As shown in Tab. 6, our Nemotron-Flash family achieves the lowest decoding latency and the best accuracy among models of comparable size. For example, Nemotron-Flash-1B delivers 5.5% higher accuracy than Qwen3-0.6B with 1.9× latency reduction and 46× higher throughput. Similarly,

Table 6: We benchmark our Nemotron-Flash-1B/3B against SOTA SLMs across 16 tasks, including MMLU, commonsense reasoning (CR), math, coding, and recall tasks. Latency is measured on an NVIDIA H100 GPU for decoding 8k tokens with a batch size of 1 using CUDA Graph. Throughput (Thr.) is measured with a 32k-token input length using the maximum batch size w/o OOM. Nemotron-Flash-3B-*TP* is a throughput-optimized variant configured with the 1FA+2SWA setting in Sec. 3.3.

| Model | Param. | Depth | BS=1 Latency (s) | Max BS Thr. (tok/s) | MMLU (%) | CR (%) | Math (%) | Coding (%) | Recall (%) | Avg. (%) |
|---|---|---|---|---|---|---|---|---|---|---|
| AMD-OLMo-1B | 1B | 16 | 24.66 | 142 | 27.11 | 51.21 | 12.66 | 7.35 | 60.32 | 35.63 |
| Llama-3.2-1B | 1.2B | 16 | 21.44 | 932 | 31.06 | 50.82 | 17.24 | 24.15 | 65.41 | 41.45 |
| Qwen2.5-0.5B | 0.5B | 24 | 22.81 | 2382 | 47.61 | 47.52 | 32.66 | 32.06 | 65.41 | 45.16 |
| Qwen3-0.6B | 0.6B | 28 | 27.55 | 160 | **52.44** | 48.91 | **36.88** | 24.32 | 62.92 | 44.11 |
| Nemotron-Flash-1B | 0.96B | 12 | **14.45** | **7289** | 44.63 | 54.46 | 34.86 | 37.91 | 67.11 | **49.63** |
| H2O-Danube3-4B | 4B | 24 | 45.92 | 178 | 53.76 | 60.79 | 34.73 | 11.65 | 58.45 | 44.37 |
| SmolLM2-1.7B | 1.7B | 24 | 31.11 | 151 | 50.21 | 58.61 | 32.88 | 21.11 | 62.42 | 46.21 |
| Llama-3.2-3B | 3B | 28 | 45.47 | 173 | 56.30 | 58.11 | 30.07 | 34.80 | 70.01 | 50.89 |
| Qwen2.5-1.5B | 1.5B | 28 | 34.50 | 687 | 60.68 | 56.29 | 48.14 | 42.95 | 69.32 | 54.65 |
| Qwen3-1.7B | 1.7B | 28 | 36.20 | 157 | 62.46 | 57.21 | 53.71 | 43.76 | 66.42 | 55.47 |
| Qwen2.5-3B | 3B | 36 | 49.40 | 459 | **65.56** | 58.88 | 53.83 | 49.46 | 73.03 | 58.96 |
| Nemotron-Flash-3B | 2.7B | 18 | 28.71 | 2939 | 61.19 | **61.02** | 57.62 | **53.33** | 73.25 | **60.98** |
| Nemotron-Flash-3B-*TP* | 2.7B | 18 | **27.95** | **4657** | 61.19 | 60.94 | **57.88** | 52.40 | **73.77** | 60.84 |

Nemotron-Flash-3B achieves +2.0%/+5.5% higher average accuracy, 1.7×/1.3× latency reduction, and 6.4×/18.7× higher throughput than Qwen2.5-3B/Qwen3-1.7B, respectively. In addition, with further optimization of the attention configuration, Nemotron-Flash-3B-*TP* achieves 10.1× and 29.7× higher throughput compared to Qwen2.5-3B and Qwen3-1.7B, respectively.

It is worth noting that (1) in addition to achieving the most competitive latency and throughput, Nemotron-Flash-3B attains the highest accuracy in commonsense reasoning, math, coding, and recall tasks among models larger than 1.5B parameters; and (2) although Nemotron-Flash-1B and Nemotron-Flash-3B contain only 2 and 3 full-attention layers, respectively, both achieve the most competitive recall accuracy, suggesting that maintaining full KV cache across all layers is unnecessary, which is consistent with observations from existing hybrid LMs [30, 40].

## 3.2 Benchmark with SOTA Instruct SLMs

We instruction-tune the Nemotron-Flash-3B model using a two-stage supervised fine-tuning (SFT) strategy on two proprietary datasets. The learning rates for the first and second stages are set to 8e-6 and 5e-6, respectively. Each stage is trained for one epoch using a cosine learning rate scheduler and a global batch size of 384. To accelerate training, we adopt the efficient packing strategy from prior work [41, 42, 43], with a block size of 29,000 tokens.

We benchmark Nemotron-Flash-3B-Instruct against Qwen2.5-1.5B and Qwen3-1.7B across MMLU (5-shot), GPQA (0-shot), GSM8K (5-shot), and IFEval. As shown in Table 7, Nemotron-Flash-3B-Instruct demonstrates strong reasoning and instruction-following capabilities, achieving the best average accuracy and efficiency, e.g., over +4.7% average accuracy and 4.3×/18.7× higher throughput compared to Qwen2.5-1.5B and Qwen3-1.7B, respectively. Despite having over 1.6× more parameters, which contribute to enhanced intelligence, Nemotron-Flash maintains superior real-device efficiency, owing to its architectural improvements.

Table 7: Benchmarking Nemotron-Flash-3B-Instruct with SOTA Instruct SLMs.

| Instruct Model | Param. | Lat. (s) | Thr. (tok/sec) | MMLU | GPQA | GSM8K | IFEval | Avg |
|---|---|---|---|---|---|---|---|---|
| SmolLM2-1.7B | 1.7B | 31.11 | 151 | 49.11 | 29.24 | 47.68 | **55.06** | 45.27 |
| Qwen2.5-1.5B | 1.5B | 34.50 | 687 | 59.73 | **30.13** | 56.03 | 46.78 | 48.17 |
| Qwen3-1.7B | 1.7B | 36.20 | 157 | 60.18 | 28.34 | 64.88 | 31.29 | 46.17 |
| Nemotron-Flash-3B | 2.7B | **28.71** | **2939** | 60.34 | 29.54 | **69.45** | 52.03 | **52.84** |

## 3.3 Ablation Study on Attention Configuration

Full attention (FA) operators are essential for long-context retrieval; however, they also become the primary bottleneck for large-batch-size long-context throughput. To better understand this trade-off, we conduct an ablation study on the attention configurations of Nemotron-Flash-3B. Starting from the pretrained Nemotron-Flash-3B base model with three FA layers, we perform continuous pretraining with a 29k context length for 25B tokens under three configurations: (1) three FA layers, (2) two FA layers plus one SWA layer with an 8k window size, and (3) one FA layer and two SWA layers.

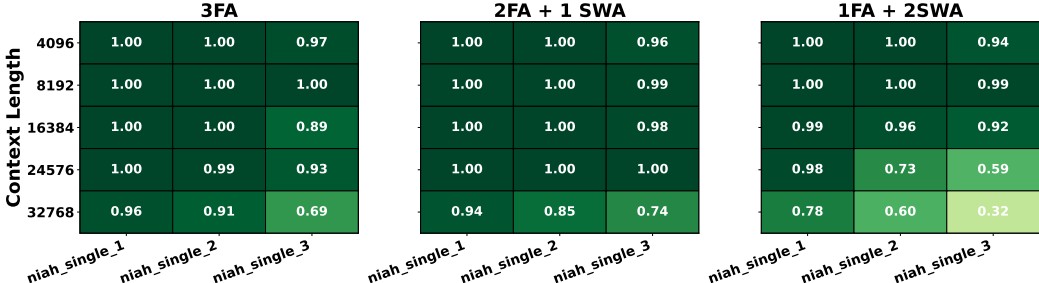

Figure 9: Visualizing the NIAH scores of different attention configurations on Ruler [44].

We report the accuracy on both general benchmarks and three needle-in-a-haystack (NIAH) tasks from Ruler [44] in Tab. 8 and Fig. 9, respectively. The results show that (1) replacing more FA layers with SWA substantially improves throughput, e.g., the 1FA+2SWA configuration achieves $1.6\times$ higher throughput than the 3FA setting; (2)

Table 8: Comparing Nemotron-Flash-3B variants with different attention configurations. Decoding throughput is measured with a 32k-token input length using the maximum batch size without OOM.

| Setting | BS=1 Lat (s) | Max BS Thr. (tok/s) | MMLU | CR | Math | Coding | Recall | Avg |
|---|---|---|---|---|---|---|---|---|
| 3FA | 28.71 | 2939 | 61.19 | 61.02 | 57.62 | 53.33 | 73.25 | 60.98 |
| 2FA+1SWA | 28.06 | 3737 | 61.02 | 60.93 | 58.56 | 51.82 | 73.37 | 60.69 |
| 1FA+2SWA | 27.95 | 4657 | 61.19 | 60.94 | 57.88 | 52.40 | 73.77 | 60.84 |

general benchmark accuracy, including recall performance, remains largely unaffected when more SWA layers with 8k window size are used; and (3) NIAH performance, however, drops significantly at longer context lengths when the number of FA layers is reduced to one, highlighting the importance of FA operators for long-context capability. Therefore, we recommend maintaining at least two full attention layers even in SLMs.

## 4 Related Work

**Small language models.** The large model size and computational demands of LLMs [45, 46, 47, 48, 49, 50, 51, 52] hinder their efficient deployment on resource-constrained platforms. This has motivated the development of SLMs, such as MobileLLM [1], MiniCPM [9], PanGu-$\pi$ Pro [11], and TinyLlama [53]. These works aim to deliver parameter-efficient SLMs within a given parameter budget. However, parameter efficiency alone often does not translate into proportional latency reductions on real devices. Our work targets real-device efficiency and aims to provide insights and methodologies for developing latency-optimal SLMs.

**Efficient attention alternatives.** To address the quadratic computation and linearly increasing memory of attention modules, efficient attention alternatives with sub-quadratic complexity in sequence length have been proposed [54, 55, 14, 15, 16, 4, 17, 18]. Notable examples include RWKV [54], RetNet [55], Mamba [14], Mamba2 [15], GLA [16], DeltaNet [17], Gated DeltaNet [18], and Jet-Block [56], each featuring different memory update rules. However, despite their potential, linear attention mechanisms have been found to exhibit limited recall capabilities [6] and to underperform standard attention mechanisms on in-context learning tasks [57].

**Hybrid language models.** To combine the efficiency of linear attention with the recall capabilities of quadratic attention, hybrid models that incorporate both types of operators have emerged. Specifically, [57, 6, 5, 22, 23] sequentially stack Mamba and attention layers in hybrid models and show enhanced commonsense reasoning and long-context capabilities. [7] proposes a hybrid-head structure that stacks attention and Mamba in parallel. However, existing hybrid models still rely on manual operator combinations, requiring tedious trial-and-error processes. Our work aims to automate operator combination in hybrid models, enabling more scalable hybrid model development.

## 5 Conclusion

This work systematically explores key architectural and training factors essential for developing latency-optimal SLMs. By analyzing optimal depth-width ratios, strategically combining efficient attention operators through an evolutionary search framework, and enhancing training with weight normalization and meta tokens, we establish a comprehensive framework that significantly improves both real-device latency and accuracy, and deliver the Nemotron-Flash model family that advances the SOTA accuracy-latency frontier. Beyond the framework, we hope the actionable insights and guidelines we provide will shed light on future research and development of low-latency / high-throughput SLMs tailored to diverse, latency-sensitive real-world applications.

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

# A  Detailed Experimental Settings

**Depth-width scaling in Sec. 2.1.** We train a series of Llama models with five depth settings: 6, 12, 18, 24, and 30 blocks, where each block contains one attention and one FFN. For each depth setting, we further vary the model width, i.e., hidden size, to create different models, ensuring that the resulting models across different depths have relatively comparable parameter ranges. We train each model on 100B tokens from the Smollm-corpus [12] using the AdamW optimizer and a cosine learning rate schedule with an initial learning rate of 5e-4.

**Architecture explorations in Sec. 2.2.** For vanilla Mamba and Mamba2 models, we do not insert FFNs, following their original papers. For all other pure models we explored, we add one FFN after each token mixing operator. For hybrid models, we adopt the building block of Operator1-Operator2-FFN. For example, when building a hybrid model with DeltaNet and Mamba2, we use the building block DeltaNet-Mamba2-FFN. All models have 500M parameters and consist of 24 operators, where each token mixing operator and each FFN is counted as one operator. We train each model on 100B tokens from the Smollm-corpus [12] using the AdamW optimizer and a cosine learning rate schedule with an initial learning rate of 5e-4. The same training settings are used for training the searched architectures, as well as for the meta token experiments in Sec. 2.4.

**Weight normalization experiments in Sec. 2.3.** For all models with and without weight normalization, we train them on 100B tokens from the Smollm-corpus [12] using the AdamW optimizer and a cosine learning rate schedule with an initial learning rate of 1e-3. This learning rate is the tuned learning rate that achieves the best convergence for models without weight normalization. We then apply weight normalization on top of this setting to demonstrate that our method can further boost task performance beyond the best baseline.

Table 9: We benchmark our Nemotron-Flash-1B/3B against SOTA SLMs across 16 tasks, including MMLU, commonsense reasoning (CR), math, coding, and recall tasks. Latency is measured on an NVIDIA H100 GPU for decoding 8k tokens with a batch size of 1 using CUDA Graph. Throughput is measured with a 32k-token input length using the maximum batch size w/o OOM.

| Model | MMLU | Commonsense Reasoning | | | | | | | Math | | | Coding | | | | | Recall | | | | Overall Avg |
|---|---|---|---|---|---|---|---|---|---|---|---|---|---|---|---|---|---|---|---|---|---|
| | | PIQA | ARCC | ARCE | Hellaswag | Winogrande | OBQA | Avg | GSM8k | MathQA | Avg | HumanEval | HumanEval-Plus | MBPP | MBPP-Plus | Avg | FDA | SWDE | Squad | Avg | |
| AMD-OLMo-1B | 27.11 | 74.92 | 31.66 | 65.95 | 47.29 | 61.64 | 25.80 | 51.21 | 1.36 | 23.95 | 12.66 | 5.49 | 6.10 | 5.40 | 12.43 | 7.35 | 70.15 | 77.05 | 33.75 | 60.32 | 35.63 |
| Llama-3.2-1B | 31.06 | 74.37 | 31.06 | 65.45 | 47.71 | 59.91 | 26.40 | 50.82 | 5.31 | 29.18 | 17.24 | 17.07 | 14.02 | 26.60 | 38.89 | 24.15 | 74.14 | 84.25 | 37.84 | 65.41 | 41.45 |
| Qwen2.5-0.5B | 47.61 | 70.29 | 29.44 | 64.44 | 40.69 | 55.49 | 24.80 | 47.52 | 36.47 | 28.84 | 32.66 | 28.05 | 25.61 | 29.60 | 44.97 | 32.06 | 68.69 | 79.39 | 48.16 | 65.41 | 45.16 |
| Qwen3-0.6B | 52.44 | 69.69 | 33.36 | 65.65 | 41.06 | 58.88 | 24.80 | 48.91 | 42.84 | 30.92 | 36.88 | 19.51 | 17.07 | 22.60 | 38.10 | 24.32 | 64.88 | 80.29 | 43.60 | 62.92 | 44.11 |
| Nemotron-Flash-1B | 44.63 | 75.41 | 41.47 | 74.83 | 45.80 | 59.67 | 29.60 | 54.46 | 32.52 | 37.19 | 34.86 | 32.93 | 29.27 | 35.20 | 54.23 | 37.91 | 77.59 | 77.23 | 46.51 | 67.11 | 49.63 |
| H2O-Danube3-4B | 53.76 | 79.60 | 46.67 | 77.19 | 58.78 | 68.51 | 34.00 | 60.79 | 41.62 | 27.84 | 34.73 | 1.83 | 1.83 | 17.00 | 25.93 | 11.65 | 33.48 | 89.20 | 52.68 | 58.45 | 44.37 |
| SmolLM2-1.7B | 50.21 | 77.20 | 44.45 | 77.74 | 53.38 | 66.06 | 32.80 | 58.61 | 31.31 | 34.44 | 32.88 | 0.61 | 0.61 | 35.60 | 47.62 | 21.11 | 59.07 | 82.18 | 46.01 | 62.42 | 46.21 |
| Llama-3.2-3B | 56.30 | 76.44 | 42.49 | 74.37 | 55.31 | 69.06 | 31.00 | 58.11 | 25.40 | 34.74 | 30.07 | 26.22 | 23.78 | 36.80 | 52.38 | 34.80 | 78.49 | 89.38 | 42.16 | 70.01 | 50.89 |
| Qwen2.5-1.5B | 60.68 | 75.79 | 41.38 | 75.72 | 50.20 | 63.06 | 31.60 | 56.29 | 61.87 | 34.41 | 48.14 | 36.59 | 31.10 | 44.60 | 59.52 | 42.95 | 71.14 | 86.14 | 50.67 | 69.32 | 54.65 |
| Qwen3-1.7B | 62.46 | 75.89 | 41.64 | 73.23 | 49.39 | 63.93 | 39.20 | 57.21 | 69.60 | 37.82 | 53.71 | 36.59 | 36.59 | 43.00 | 55.82 | 43.76 | 65.43 | 85.33 | 48.49 | 66.42 | 55.47 |
| Qwen2.5-3B | 65.56 | 78.40 | 44.80 | 77.31 | 54.87 | 68.51 | 29.40 | 58.88 | 70.43 | 37.22 | 53.83 | 37.20 | 31.71 | 56.20 | 72.75 | 49.46 | 72.78 | 90.37 | 55.93 | 73.03 | 58.96 |
| Nemotron-Flash-3B | 61.19 | 79.65 | 50.17 | 80.72 | 53.79 | 67.01 | 34.80 | 61.02 | 66.34 | 48.91 | 57.62 | 48.17 | 43.29 | 52.00 | 69.84 | 53.33 | 81.85 | 85.42 | 52.48 | 73.25 | 60.98 |

**Tasks for computing average commonsense reasoning (CR) accuracy.** Unless otherwise specified, in the above exploration experiments, we adopt eight tasks to compute the average CR accuracy: Lambda, PIQA, ARC-Easy, ARC-Challenge, Hellaswag, Winogrande, TruthfulQA, and SIQA, all from the lm-evaluation-harness [39].

# B   More Benchmark Results of Nemotron-Flash

As a complement to the domain-averaged accuracies reported in Sec. 3.1 and Tab. 6, we present the detailed per-task accuracies within each domain in Tab. 9.

# C   Evolutionary Search: More Details and Experiments

## C.1   Detailed Search Space

**Search template.** We divide the entire architecture into three stages, with each stage repeating one type of building block. This three-stage strategy balances operator heterogeneity with architectural regularity. We search for both the number of blocks in each stage and the structure of each building block, including the involved operators, their ratios, and the ratio of FFNs.

**Operators and their ratios.** Based on the exploration of pure and hybrid models in Sec. 2.2, we include attention, Mamba2, and DeltaNet in our search space, considering both operator efficiency and their complementary roles. To ensure regularity and avoid overly complex blocks, we allow at most two types of operators (excluding FFNs) in each building block type. We support different ratios between two hybrid operators: 0:1 (i.e., only one operator), 1:1, 1:2, and 1:3.

**FFN ratios.** We allow for 0, 1, or 2 FFNs after each token mixing operator in each building block.

**Number of building blocks.** We allow flexible configurations of the number of building blocks in each stage, as long as the total number of operators does not exceed a predefined limit (i.e., 30). If the total exceeds this limit, we reduce the number of building blocks in the last stage to meet the depth constraint.

After determining all the above design factors for a new architecture, we select a hidden size from [1024, 1280, 1536, 1792, 2048, 2304, 2560], choosing the largest size that satisfies the target latency. This strategy allows us to use the short-training PPL solely to rank the searched architectures.

## C.2   Our Evolutionary Search Algorithm

As a complement to the search algorithm described in Sec. 2.2, we present our adopted aging evolutionary search in Alg. 1, which iteratively refines a population of hybrid language model architectures by selectively sampling and mutating architectures over multiple cycles.

**Search process.** After initializing a population of architectures from a set of seeded and randomly initialized architectures, a set of candidate architectures (we use 10) is sampled, mutated, and evaluated concurrently in each cycle. Mutations are performed by altering one of the searchable factors: (1) changing one operator in a building block type, (2) varying the hybrid operator ratio in a building block type, (3) adjusting the FFN ratio in a building block type, or (4) modifying the number of building blocks across stages. The sampled architectures are evaluated using short-training perplexity as an efficient proxy for final performance, and latency is quickly estimated using a pre-

**Algorithm 1:** Aging Evolutionary Search for Hybrid Operator Combinations

---

**Input:** LUT for latency, population size $P$, sample size $S$, number of cycles $C$, target latency budget $L_{target}$

**Output:** Best-performing architecture

1  Initialize population $\mathcal{P} \leftarrow$ Seeded architectures;
2  Short-train each architecture in $\mathcal{P}$ and compute their proxy PPL and latency;
3  **for** $cycle \leftarrow 1$ **to** $C$ **do**
4     **for** *each architecture in current cycle* **in parallel** **do**
5        Sample subset $\mathcal{S} \subseteq \mathcal{P}$ randomly, with $|\mathcal{S}| = S$;
6        Select the parent $p \leftarrow \arg\min_{x \in \mathcal{S}} \mathrm{PPL}(x)$ s.t. latency$(x) \leq L_{target}$;
7        Create offspring $c$ by mutating one factor of $p$ (operator ratios, FFN ratios, or block type counts), adjusting to maintain depth constraint;
8        Evaluate $c$ by short-training to obtain proxy PPL and querying LUT for latency;
9        Remove oldest architecture from $\mathcal{P}$;
10       Add offspring $c$ to population $\mathcal{P}$;
11 **return** architecture in $\mathcal{P}$ with lowest proxy PPL satisfying latency constraint $L_{target}$;

---

measured lookup table (LUT). This parallel, proxy-driven strategy effectively balances the exploration of novel designs and the exploitation of known strong performers, facilitating rapid convergence to high-quality architectures.

**Short training settings.** We train each model on 10B tokens from the Smollm-corpus [12] using the AdamW optimizer and a cosine learning rate schedule with an initial learning rate of 5e-4. Training and evaluating a sampled architecture takes approximately 2 hours using 32 NVIDIA A100 GPUs.

## C.3  Search with Parameter Efficiency as the Objective

As a complement to the searched architecture optimized for parameter efficiency, presented in Sec. 2.2 and Tab. 3, we provide a benchmark against other baseline architectures, all with 500M parameters, in terms of Wikitext PPL and CR accuracy averaged over eight tasks. All models are trained on 100B tokens from the Smollm-corpus [12] using the AdamW optimizer and a cosine learning rate schedule with an initial learning rate of 5e-4. As shown in Tab. 10, the searched architecture achieves over 1.21% higher CR accuracy and a reduction of more than 0.74 in PPL compared to all 500M baselines.

Table 10: Benchmarking our searched architecture, optimized for parameter efficiency, against baseline architectures. All models have 500M parameters. CR accuracy is averaged over eight tasks.

| Metric | Attention | GLA | DeltaNet | Gated DeltaNet | RWKV | Mamba | Mamba2 | Searched (Ours) |
|--------|-----------|-----|----------|----------------|------|-------|--------|-----------------|
| Wiki PPL | 24.44 | 25.16 | 23.87 | 23.80 | 25.47 | 24.55 | 24.36 | **23.06** |
| CR Acc (%) | 48.02 | 46.82 | 47.83 | 47.96 | 46.46 | 47.32 | 47.72 | **49.23** |

## C.4  Configurations of Our Final Nemotron-Flash Models

We provide the detailed configurations of our final Nemotron-Flash models in Tab. 11.

Table 11: Architectural details of our Nemotron-Flash models. M2, A, D, and F denote Mamba2, attention, DeltaNet, and FFN, respectively. For attention layers, we use group-query attention with a group size of 4, with the specific number of heads detailed in the table.

| Model | Hidden Size | FFN Dim | Attn/KV Heads | #Meta Tokens | #Operators | Operators |
|-------|-------------|---------|---------------|--------------|------------|-----------|
| Nemotron-Flash-1B | 2048 | 6144 | 16/4 | 256 | 24 | [D, F, M2, F, A, F, M2, F, D, F, M2, F, A, F, M2, F, D, F, M2, F, D, F, M2, F] |
| Nemotron-Flash-3B | 3072 | 9216 | 24/6 | 256 | 36 | [D, F, M2, F, A, F, M2, F, D, F, M2, F, A, F, M2, F, D, F, M2, F, A, F, M2, F, D, F, M2, F, D, F, M2, F, D, F, M2, F] |

Table 12: Comparing the achieved task accuracy and training speed using vanilla models, weight normalization, nGPT without weight normalization, and full nGPT.

| Model | Setting | Wiki PPL | CR Acc (%) | Training Time (sec.) / Iter |
|---|---|---|---|---|
| LLaMA 1B | vanilla | 18.67 | 53.81 | 0.43 |
| | wnorm | 18.03 | **54.85** | 0.46 |
| | nGPT w/o wnorm | 19.38 | 51.83 | 0.56 |
| | nGPT | **17.45** | 54.73 | 0.59 |
| Hybrid 1B | vanilla | 17.96 | 54.36 | 0.78 |
| | wnorm | 17.50 | **55.74** | 0.81 |
| | nGPT w/o wnorm | 18.96 | 52.44 | 1.01 |
| | nGPT | **17.25** | 55.33 | 1.04 |

# D    Weight Normalization: More Analysis and Experiments

## D.1    Compare Weight Normalization with nGPT

As discussed in Sec. 2.3, our weight normalization technique can be viewed as a simplified and efficient variant of nGPT [28], which enforces all computations in LMs to operate on a unit sphere by introducing multiple activation normalization layers in addition to weight normalization. To analyze the performance breakdown achieved by nGPT, we evaluate our weight normalization, nGPT without weight normalization (i.e., activation normalization only), and full nGPT on two models: a Llama 1B model and a 1B hybrid model with each building block structured as Attention-Mamba-FFN.

As shown in Tab. 12, we observe that: (1) Using only our weight normalization achieves strong task performance, with higher CR accuracy than nGPT and slightly worse language modeling PPL; (2) Weight normalization is essential to nGPT, as removing it leads to a notable drop in accuracy; (3) The additional activation normalization layers in nGPT introduce over 30% training overhead, reducing the number of training tokens under a fixed training budget. Thus, our weight normalization offers a more efficient alternative to nGPT while maintaining comparable task performance.

## D.2    Gradient Norm Evolution of More Models

As a complement to Sec. 2.3, we further visualize the gradient norm evolution during training with and without weight normalization for four models in Fig. 10. We observe that weight normalization results in a slightly increased gradient norm, leading to larger relative weight updates, considering the smaller weight norm. This effect is particularly beneficial in the later stages of training and leads to consistent final accuracy improvements across model families.

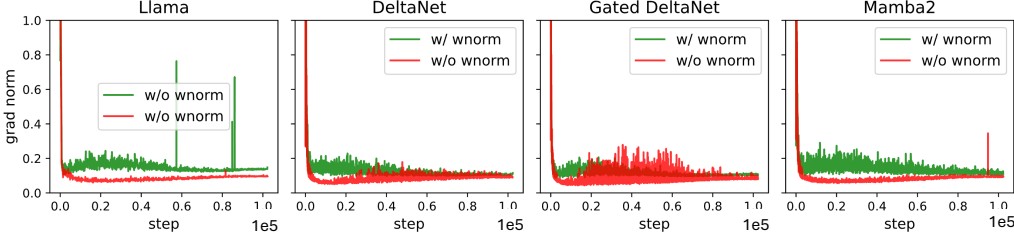

Figure 10:  The gradient norm evolution during training w/ and w/o weight normalization.

# E    The Choice of Tokenizers

Previous parameter-efficient SLMs [1] adopt a tokenizer with a small vocabulary size, such as Llama2 [47], to reduce the number of parameters. In contrast, we use a tokenizer with a larger vocabulary size, i.e., Mistral-NeMo-Minitron [30]. The rationale is that a larger vocabulary often results in more concise token representations and fewer tokens when encoding the same sentence. For example, the phrase by the way" can be tokenized into a single by-the-way" token.

As shown in Tab. 13, when averaging the number of tokens per sample over two datasets, the Mistral-NeMo-Minitron tokenizer achieves a 9.3% and 13.5% reduction in token count. At the same time, the decoding latency for generating 8k tokens using models with embedding dimensions corresponding to the two tokenizers shows

Table 13: Tokens per sample on AG News / Wikitext datasets using LLaMA2 [47] and Mistral-NeMo-Minitron [30] tokenizers.

| Dataset | LLaMA2 | Mistral-NeMo-Minitron |
|---------|--------|----------------------|
| AG News | 63.64 | 55.08 (-13.5%) |
| Wikitext | 116.86 | 106.00 (-9.3%) |

only a 5.8% latency gap (38.93 seconds vs. 41.31 seconds). This suggests that overall efficiency could be higher when using tokenizers with larger vocabularies. Furthermore, given that larger vocabulary sizes often improve task performance, we adopt the Mistral-NeMo-Minitron tokenizer [30] over smaller alternatives.

## F   Deployment Flow

Existing deployment frameworks such as TensorRT-LLM [58] and vLLM [59] support Attention and Mamba2, while linear attention variants like DeltaNet are only supported by FlashLinearAttention [21]. Hybrid LMs involve diverse operators and require a deployment flow that can efficiently accelerate each operator type. To this end, we integrate TensorRT-LLM's AutoDeploy kernels [10] with efficient KV cache management for full Attention, the official implementation of Mamba2 [15], and DeltaNet via FlashLinearAttention [21] to deploy our Nemotron-Flash. The entire model is wrapped with CUDA Graph to mitigate kernel launch overhead, which is critical for edge applications with a batch size of 1.

To demonstrate the effectiveness of our deployment flow, we benchmark the decoding speed of a Llama model (12 blocks, hidden size of 1536, 550M parameters) using PyTorch, vLLM, TensorRT-LLM, and our deployment flow. As shown in Tab. 14, our deployment flow outperforms vLLM in decoding speed at longer sequence lengths and achieves a latency gap within 15% of TensorRT-LLM for full-attention models. This gap could be further reduced with improved full-attention kernel optimization on top of [10]. Additionally, our deployment supports linear attention variants, which are not supported by vLLM or TensorRT-LLM. This set of comparisons validates the effectiveness of our deployment flow.

Table 14: Decoding latency (sec.) at various decoding lengths for a 550M Llama model across different deployment flows. The batch size is set to 1, as our target is edge applications.

| Deployment Flow | 6 | 64 | 256 | 512 | 1024 | 2048 | 4096 | 8192 | 16384 | 32768 |
|-----------------|------|------|------|------|-------|-------|-------|--------|--------|--------|
| PyTorch | 0.16 | 0.96 | 3.64 | 7.07 | 14.28 | 28.89 | 56.92 | 113.93 | 222.78 | 457.72 |
| vLLM | 0.02 | 0.14 | 0.55 | 1.09 | 2.16 | 4.31 | 8.66 | 17.82 | 67.23 | 169.74 |
| TensorRT-LLM | 0.17 | 0.30 | 0.64 | 1.00 | 1.81 | 3.42 | 6.72 | 13.65 | 28.41 | 60.41 |
| Ours | 0.01 | 0.14 | 0.54 | 1.08 | 2.17 | 4.36 | 8.72 | 17.54 | 35.29 | 71.65 |

## G   Limitations and Future Work

In our architecture search for SLMs, we primarily use language modeling as the search proxy, which may not fully capture other capabilities such as long-context understanding. Additionally, our search focuses on macro-architecture, leaving room to explore more fine-grained design factors. We plan to extend our search framework to optimize for these aspects in future work.

## H   Societal Impact

Our work aims to develop latency-optimal SLMs that lower the barrier to LM deployment and facilitate ubiquitous edge intelligence. A potential negative societal impact is that the broader use of LMs may increase the risk of misuse, emphasizing the need for more advanced and efficient safety enhancement techniques.

