# OpenReview forum: "Nemotron-Flash: Towards Latency-Optimal Hybrid Small Language Models"
_NeurIPS.cc/2025/Conference — NeurIPS 2025 poster_

### Official Review · Reviewer_ictF · 2025-07-02

**Clarity:** 2
**Significance:** 2
**Originality:** 3
**Rating:** 4
**Confidence:** 4

**Summary:**

This paper introduces Fast-SLM, a new series of small language models designed to run faster on real devices, not just look small in size. The authors point out that having fewer parameters doesn’t always mean the model will be faster in practice.

To solve this, they propose a full strategy that considers both how the model is built and how it's trained, aiming for the best balance between speed and accuracy.

Their key ideas include:

1. Model Structure: They study how deep and wide a model should be. Instead of blindly following the trend of "deep and narrow" models, they use a math-based method to find the best structure for different speed limits.
2. Operator Mixing: They test various attention mechanisms (like Mamba2, DeltaNet, etc.) and show that mixing them in one model works better than using just one type. They even use automated search to find the best mix.
3. Training Tricks: They add two smart tricks:
     - Weight normalization for more stable and effective training.
     - Meta tokens to speed up generation and improve accuracy.
4. Fast-SLM Models: They build two models (960M and 2.7B) using the above ideas. These models beat top models like Llama3 and Qwen3 in both speed and accuracy. For example, Fast-SLM-2.7B is 1.46× faster, 2.57% more accurate, and uses 10× less memory than Llama-3.2-3B.
5. Real-world Testing: They test the models on 9 language tasks and show that Fast-SLM runs fast and performs well using real deployment tools like TensorRT-LLM and CUDA Graph.

**Questions:**

Questions and Suggestions for the Authors
1. How much of the latency improvement comes from the model design vs. deployment tools (e.g., CUDA Graph, TensorRT)?
You report large speedups, but it’s unclear how much is due to architectural changes versus low-level software optimization.
Could you run an ablation showing performance without CUDA Graph or AutoDeploy, to understand what is “portable” across platforms like CPU or non-NVIDIA chips? This would increase confidence in the general usefulness of Fast-SLM.

2. Can your hybrid operator search framework be used under realistic training budgets (e.g., <50B tokens, <32 GPUs)?
The search method looks great, but you trained on 100B tokens or more. That’s hard to reproduce.
Can you demonstrate that this search method is still effective with limited resources, like a smaller dataset or fewer GPUs? If yes, it would make your method much more appealing to both academia and small-scale industry.

3. Why no evaluation on long-context or multi-turn tasks?
You emphasize latency, but reducing cache may hurt long-context understanding or dialogue.
Can you report performance on long-context benchmarks (like L-Eval, LongBench, or multi-turn QA)? If results are strong, it will show Fast-SLM is not just fast, but also broadly capable.

4. Have you considered risks of easy on-device misuse (e.g., spam bots, data leakage)?
Fast, on-device models are great—but also make LLM misuse cheaper and harder to monitor.
Do you plan to release safety filters or usage restrictions? What’s your view on responsible deployment? A short note on this would help align with responsible AI practices, especially for industrial readers.

5. Could your scaling law for depth-width trade-off be released as a public tool?
Your method for selecting the best depth-width combo seems useful beyond this paper.
Would you consider releasing a simple API or table that helps others design latency-efficient models on their own hardware? This would increase the paper’s impact and adoption in the community.

**Ethical Concerns:**

["NO or VERY MINOR ethics concerns only"]

**Limitations:**

Limitations:

A few important and practical limitations could be discussed more explicitly:

1. Deployment Dependency: The reported speedups depend heavily on deployment tricks like TensorRT-LLM AutoDeploy and CUDA Graph. These are great tools—but not every user or platform (e.g., CPU-only devices, non-NVIDIA hardware) can use them. The authors could clarify how much speedup truly comes from their architecture, and how portable the results are.
2. Training Cost and Accessibility: The paper mentions training on 1.6T–2.2T tokens using 256×H100 GPUs. This scale is out of reach for most researchers and companies. It would be helpful to discuss whether the benefits of Fast-SLM (e.g., search, hybrid design) still hold under smaller budgets.
3. Underexplored Risks in Cache Reduction: While cache size is drastically reduced (which is good), aggressively minimizing cache might affect context consistency or in-context learning ability in real-world usage. This trade-off deserves more discussion.

**Quality:**

2

**Strengths And Weaknesses:**

Strengths
1. Clear Motivation: The paper clearly explains why current small models aren't always fast on real devices, and why optimizing only for parameter count isn't enough.
2. Practical Significance: The focus on real-device latency (instead of just model size) makes this work very relevant for real-world deployment (e.g., mobile, edge, low-latency applications).
3. Useful Insights:The depth-width trade-off and operator mixing analysis are well presented and offer general guidelines for future model design.
4. Strong Results: The Fast-SLM models clearly outperform popular models like LLaMA and Qwen in both speed and accuracy. The 10× reduction in cache usage is especially impressive.
5. Simple Training Tricks with Real Impact: The proposed weight normalization and meta tokens are lightweight, easy to use, and give noticeable gains.

Weaknesses

1. Limited Novelty in Techniques: Many of the components (e.g., scaling laws, hybrid models, weight norm, meta tokens) are based on existing ideas. The novelty comes more from combining them well rather than introducing fundamentally new methods.
2. Lack of Statistical Significance Analysis: The paper does not provide error bars or variance across runs. That makes it harder to judge whether the improvements are robust.
3. Search Focus Is Narrow And Searching Efficiency Is Low: The architecture search only considers macro-level blocks (e.g., which operator to use), but does not explore fine-grained design choices like number of heads, expansion ratios, etc.
Limited Task Diversity: All benchmarks focus on commonsense reasoning and language modeling. There is no evaluation on longer-context tasks, dialog, or code, which could show broader generalization.
4. Deployment Tuning Details Missing: While real-device latency is central to the paper, deployment tricks (e.g., CUDA Graph, TensorRT) might play a large role in the speedups. It’s not always clear how much of the gain comes from architecture vs. engineering.

---

> ### Author Rebuttal · Authors · 2025-07-31
>
> Thank you for recognizing the “useful insights,” “strong results,” and “practical significance” of our work, as well as for your constructive comments! We have addressed all of your questions below:
>
> **1. Novelty of our techniques**
>
> We respectfully emphasize that the key contribution of this work lies in identifying critical factors affecting SLM real-device latency and offering generalizable insights and methodologies for latency-aware SLM design and training.
>
> Specifically, while depth–width ratios and operator choices are known design factors, their impact on real-device latency and how to make effective trade-offs remain unclear, which are addressed by this work. Additionally, our weight normalization eliminates the training and inference overhead of nGPT (citation [28]) while achieving comparable or better accuracy, as analyzed in Section 2.3 of the main paper and Section 6.1 of the supplementary material.
>
> As Reviewer ictF also noted, our work is “a decent attempt towards latency-optimal SLM” and has the potential to “produce significant community impact.”
>
> ---
>
> **2. Variance across runs**
>
> Since we adopt ≥100B token training for all experiments that report accuracy, we find that the results are stable across runs. For example, we rerun the experiments in Table 4 of our manuscript three times and show all results below: the PPL variation across runs is within 0.1, and the accuracy difference is within 0.16%, leading to consistent conclusions. Therefore, we didn’t include error bars in the manuscript.
>
> ||Run1||Run2||Run3||
> |---|:---:|:---:|:---:|:---:|:---:|:---:|
> |Model|Wiki PPL|Acc (%)|Wiki PPL|Acc (%)|Wiki PPL|Acc (%)|
> |Llama 1B|18.67|53.81|18.77|53.92|18.69|53.78|
> |Llama 1B + wnorm|18.03|54.85|18.06|54.80|18.03|54.80|
> |DeltaNet 1B|18.86|53.46|18.98|53.59|18.89|53.43|
> |DeltaNet 1B + wnorm|18.19|54.39|18.20|54.43|18.23|54.50|
> |Mamba2 1B|18.44|53.3|18.52|53.44|18.53|53.37|
> |Mamba2 1B + wnorm|17.88|54.71|17.84|54.55|17.82|54.68|
>
> *Note:* Acc is averaged over 8 commonsense reasoning tasks.
>
> ---
>
> **3. Clarifications on the search cost and the search focus**
>
> For the search cost, we clarify that each candidate architecture is trained for only **10B tokens**, which takes <4 hours using 16 A100 GPUs, as mentioned in **Section 5.2 of our supplementary material**. This follows the “realistic training budgets” you suggested. (The 100B token setting you referred to applies only to the architecture ablation in Tables 1 and 3 of our manuscript, not to the search process.)
>
> Regarding the search focus, our goal is to use evolutionary search to extend manual exploration of hybrid operators into an automated pipeline for obtaining more insights on LM operator combinations, rather than to build a comprehensive NAS framework. While our framework can support finer-grained searches with minimal changes, we deliberately avoid irregular architectures (e.g., with layerwise-varying #heads) as they may complicate deployment. Following your suggestion, we will explore larger design spaces in future work.
>
> ---
>
> **4. Benchmarks on more diverse tasks: Math/Coding/Long Context**
>
> Thank you for the question! We have added more benchmarks:
>
> **(a)** Math / Coding
>
> Since the paper submission, we have continued pretraining our models on an additional 2T tokens and benchmarked against baselines across 14 tasks, including Commonsense Reasoning (MMLU, Lambda, PIQA, ARCC, ARCE, HellaSwag, Winogrande, OBQA), Math (GSM8k, MathQA), and Coding (HumanEval, HumanEval-Plus, MBPP, MBPP-Plus).
>
> As shown below, **our Fast-SLM-2.7B achieves SOTA average accuracy among all models**. Specifically, Fast-SLM-2.7B outperforms LLaMA3.2-3B/Qwen2.5-3B by 9.24%/0.90% in average accuracy with 1.46x/1.67x speed-ups, respectively. We will include these updated results in the final version.
>
> |Model|Param.|Dec. Latency (s)|Cache Size (MB)|CR (%)|Math (%)|Coding (%)|Avg (%)|
> |:---:|:---:|:---:|:---:|:---:|:---:|:---:|:---:|
> |AMD-OLMo-1B|1B|28.39|1049|49.21|12.66|7.35|32.03|
> |Llama-3.2-1B|1.2B|28.37|262|49.73|17.24|24.15|37.78|
> |Qwen2.5-0.5B|0.5B|28.19|98|48.13|32.66|32.06|41.33|
> |Qwen3-0.6B|0.6B|33.91|918|49.92|**36.88**|24.32|40.74|
> |Fast-SLM-960M (Ours)|0.96B|**19.36**|**36**|**52.54**|24.57|**38.44**|**44.51**|
> |h2o-danube3-4b-base|4B|61.94|737|60.85|34.73|11.65|43.06|
> |SmolLM2-1.7B|1.7B|41.55|1573|58.64|32.88|21.11|44.24|
> |Llama-3.2-3B|3B|60.38|918|59.37|30.07|34.80|48.16|
> |Qwen2.5-1.5B|1.5B|44.06|229|57.53|48.14|42.95|52.02|
> |Qwen3-1.7B|1.7B|46.39|918|58.62|53.71|43.76|53.67|
> |Qwen2.5-3B|3B|69.19|295|60.69|**53.83**|49.46|56.50|
> |Fast-SLM-2.7B  (Ours)|2.7B|**41.31**|**80**|**61.62**|48.88|**53.22**|**57.40**|
>
>
> **(b)** Long-context
>
> To evaluate long-context tasks, we finetune Fast-SLM-2.7B (pretrained with a 4k sequence length) on 15B tokens using a 24k sequence length. We evaluate on the Needle-In-A-Haystack (NIAH) task from RULER, using 4k~32k input lengths. For reference, we also include results of h2o-danube3-4b (pretrained with an 8k sequence length). Both models use the NTK-based RoPE extension.
>
> |Input Length|Fast-SLM-2.7B|h2o-danube3-4b-base|
> |:---:|:---:|:---:|
> |4k|0.99|0.99|
> |8k|0.97|0.97|
> |16k|1|0.93|
> |24k|0.96|0.30|
> |32k|0.81|0.79|
>
> As shown above, we observe that:
>
> (1) Fast-SLM-2.7B achieves strong NIAH scores, including reasonably good performance at the 32k input length;
>
> (2) Compared to the pure Transformer model h2o-danube3-4b, Fast-SLM achieves better scores on inputs ≥16k in length. These results highlight the promise of Fast-SLMs in handling long-context inputs. A fair comparison with more models requires aligning long-context training settings and we will include evaluations on more long-context benchmarks in the final version, given more time.
>
> We also refer to recent large hybrid LMs, e.g., Nemotron-H, MiniMax-M1, and Jamba 1.6, that replace attention with efficient linear alternatives, significantly reducing cache usage while maintaining strong long-context performance. This underscores the potential of hybrid models for long-context tasks.
>
> ---
>
> **5. Ablation study on deployment settings**
>
> We first clarify that CUDA Graph has become a standard component in nearly all popular LM deployment pipelines (e.g., TensorRT-LLM, vLLM, SGLang). This is because LMs are significantly memory-bound during decoding, and without CUDA Graph, CPU overhead can dominate the latency. **CUDA Graph is indispensable for eliminating CPU overhead and ensuring reliable profiling of LM decoding.**
>
> Additionally, **we integrated TensorRT-LLM’s attention kernel to enable CUDA Graph support for attention operators**, as the vanilla attention from HuggingFace Transformers cannot be captured by CUDA Graph. The reason we did not directly adopt TensorRT-LLM is that it does not support linear attention like DeltaNet. Therefore, we combined these components to build a unified decoding pipeline capable of running hybrid models with arbitrary operators using CUDA Graph. **This infrastructure is also a minor contribution of our work to the community.**
>
> Furthermore, following your suggestion, we have added an ablation study for each component:
>
> |Profile Setting|bs=1, out_len=8k|bs=1, out_len=8k|bs=1, out_len=8k|bs=1, out_len=32k|bs=32, out_len=32k|
> |:---:|:---:|:---:|:---:|:---:|:---:|
> |Deployment Setting|Vanilla|+ TRTLLM Kernel|+ TRTLLM Kernel + CUDA graph|+ TRTLLM Kernel + CUDA graph|+ TRTLLM Kernel + CUDA graph|
> |Qwen2.5-3B|390.32|379.08|69.19|289.70|817.46|
> |Fast-SLM-2.7B|296.32|292.63|41.31|166.82|357.19|
> |Speed-up|1.32x|1.30x|1.67x|1.74x|2.29x|
>
>
> The table shows that:
>
> (1) The use of CUDA Graph can lead to 5x~7x gaps in decoding latency due to CPU overhead. This indicates that decoding with dominant CPU overhead, especially when it can be eliminated, is not a reliable profiling setting for GPU platforms;
>
> (2) Without CUDA Graph, the latency reduction of Fast-SLM is smaller, as linear attention incurs more extensive CPU workloads than standard attention (according to github issues in the official Mamba repo), further amplifying CPU overhead.
>
> (3) Further increasing the generation sequence length or batch size can lead to even greater speed-ups for our models.
>
> Regarding non-GPU platforms, they fall outside the scope of this work. However, readers may refer to the analysis in our work to profile their target platforms and derive device-specific SLMs accordingly.
>
> ---
>
> **6. Discussions on safety issues with SLMs**
>
> Thank you for the suggestion! First, prior to the public release, we will evaluate our model on safety benchmarks, including the Aegis AI Content Safety Dataset and Garak, to ensure its safety. Second, and more broadly, we agree that on-device SLMs lower the barrier to misuse and require more safety-aware tuning to ensure reliable deployment. We will include this discussion in the final version.
>
> ---
>
> **7. Tool release for the depth-width scaling law**
>
> We will release the codebase for deriving the depth–width scaling law, along with our search pipeline and deployment workflow. We hope these tools will help the community develop SLMs tailored to their specific needs.
>
> ---
>
> **8. Benefits of Fast-SLM under small budgets**
>
> The benefits of Fast-SLM hold even under smaller training budgets. As shown in Tables 3/5 of our manuscript and in the ablation study provided to Reviewer eYGu, Fast-SLM and its individual components improve accuracy or efficiency with ≤100B tokens.
>
> Moreover, Fast-SLM’s use of efficient operators enables faster training and fine-tuning, making it well-suited for dedicated downstream tasks.
>
> ---
>
> **9. Other potential risks in cache reduction**
>
> As noted in Bullet 4 of our response, Fast-SLM achieves higher accuracy across diverse tasks (e.g., commonsense reasoning/coding) compared to baselines, and maintains reasonably good long-context performance. This suggests that, for common downstream tasks, cache reduction does not hinder accuracy. Similar trends are seen in recent large hybrid LMs, e.g., Nemotron-H, MiniMax-M1, and Jamba 1.6.

---

> ### Comment · Area_Chair_54jA · 2025-08-05
> **Dear reviewer, please read the rebuttal and participate in discussions with authors. Thank you**
>
> Dear reviewer, please read the rebuttal and participate in discussions with authors. Thank you

---

### Official Review · Reviewer_X2pZ · 2025-07-02

**Clarity:** 3
**Significance:** 3
**Originality:** 2
**Rating:** 5
**Confidence:** 3

**Summary:**

This paper focuses on deriving latency-optimal small language models (SLM). The authors first derive a scaling law of validation loss w.r.t. depth & width and find a sweet spot of the depth-width ratio that achieves the best result given a certain latency budget. Then the authors conduct an evolutionary search on hybrid architecture to find the best architecture that achieves the best tradeoff between short-term PPL and decoding latency. Then the authors employ a few more tricks: weight normalization, trainable meta tokens, larger vocab size, etc. and train a new hybrid model family Fast-SLM with 1B and 7B scale, with 3-phase training: phase 1 on 1.6T pre-training, phase 2 annealing on 600B high-quality data, and optional phase 3 with distillation.

**Questions:**

How does the LUT of the decoding latency is derived in the hybrid operator evolutionary search process? I am assuming we are not recording on all possible decoding latency and there exists some decoding latency estimated formulas for each operator. It would be helpful to include this calculation process and formula in the Appendix.

**Ethical Concerns:**

["NO or VERY MINOR ethics concerns only"]

**Final Justification:**

I believe this paper has a clear community interest and impact so I would recommend acceptance. However, I still believe there are some technical details need to be improved, but this does not affect my overall assessment.

**Limitations:**

Yes

**Paper Formatting Concerns:**

No paper formatting issues founded.

**Quality:**

3

**Strengths And Weaknesses:**

Strengths:

- (**major**) This is a decent attempt towards latency-optimal SLM and has profound implication for resource-constrained device settings, and even draft model for speculative decoding, etc. I perceive this paper will produce significant community impact if the model weights are released and the complete training recipe (codebase and the data) are open-sourced.

- The experiments and evaluation are comprehensive and they show Fast-SLM can match frontier SLM model's performance while having shorter decoding latency.

- The paper is generally clear and well-written, and provides a detailed discussion on design choices that could produce frontier SLM.

Weakness:

The paper is generally clear and has profound implications, but I believe there are a few technical details to be fixed / addressed for the rigor of study.

- (**critical**) The authors first explore the best sweet spot of depth-width ratio in **Llama models** (Figure 3), and then they explore the best **hybrid operators** that achieves the desired decoding latency-performance frontier (Table 2). Then, both designs are combined in Fast-SLM model family for training. In this case, we assume the best depth-width ratio derived in Llama models can be directly transferred to hybrid models. This is a rather strong assumption because different operators would have different dependencies over depth and width for both validation loss and latency. I know the search space would be too large if we search the scaling law for every hybrid operator combinations, **but at least we should search for the latency-val loss tradeoff on the final searched architectures directly instead of directly adapting the Llama model's scaling result.**

- I don't know if the final decoding-latency optimized architecture in Table 2 is included in Figure 5 because points at 17.7 decoding latency in Table 2 clearly does not achieve the lowest short-training PPL in Figure 5.

- **Need reconciliation**: there are some previous scaling laws (e.g. Kaplan. Chinchilla doesn't study it) suggest that the model aspect ratio's impact on the validation loss is less important than the total number of parameters (Kaplan Figure 6). **The left subfigure in Figure 2 in the paper suggests the opposite as the gap across different aspect ratio could be more significant than the gap across number of parameters.** I expect the aspect ratio to be less important for LLM but maybe more important for SLM, but we certainly need such discussion & justification in the paper.

I will retain my score at this moment.

---

> ### Author Rebuttal · Authors · 2025-07-31
>
> Thank you for describing our work as “*a decent attempt towards latency-optimal SLM*” and recognizing its potential to “*produce significant community impact*,” as well as for your constructive comments! We will open-source all of our search/training recipes and final models upon acceptance. We have addressed all of your questions below:
>
> **1. Search depth-width ratios on Llama and transfer to hybrid models**
>
> *(a) The reason for decoupling depth–width ratio search from operator search*
>
> Thank you for the question! First, we agree that jointly searching both operator combinations and depth–width ratios has greater potential to yield a strong architecture, and our codebase can support this without any modification. That said, we currently decouple depth–width ratio search from operator search for two main reasons: (1) We aim to highlight the insights from each design factor independently, with the other held fixed, so that the resulting findings can better benefit the community and inspire future work; (2) As you also noted, a large design space can make the search more difficult and time-consuming to converge.
>
> Although this work focuses on providing insights into both design factors rather than proposing a NAS framework, we agree that a joint search has greater potential and will try it for future work given more time after the rebuttal period.
>
> *(b) The scaling law derived from Llama provides good starting points for selecting the depth–width ratio of hybrid models*
>
> Additionally, we would like to mention that the scaling law derived from Llama serves as a useful guide for selecting the depth–width ratio of hybrid models. In practice, we first build the scaling law using Llama models and then use it to predict promising candidates for our hybrid models, selecting the best one based on profiling results.
>
> Specifically, the Llama-derived scaling law suggests that our Fast-SLM-960M should use 12 blocks, where each block includes one (linear) attention and one FFN. We benchmark Fast-SLM-960M-12b (our final adopted configuration) against its 8-block and 16-block variants under an iso-latency setting, i.e., all models are scaled to a comparable latency by adjusting the width.
>
> As shown in the table below, Fast-SLM 12-block achieves the best PPL and accuracy across all three settings, suggesting that the depth–width ratio predicted by the LLaMA scaling law provides a strong starting point. Note that our goal with the scaling law is not to predict a single best setting, but rather to (1) emphasize the general trend of this efficiency-critical design factor, and (2) quickly obtain a near-optimal depth–width ratio under a target latency as a good starting point.
>
> | Model | Latency (sec.) | Param | Wiki PPL | CR Acc (%) |
> |:---:|:---:|:---:|:---:|:---:|
> | Fast-SLM 8-block | 19.5 | 1211M | 20.59 | 50.21 |
> | Fast-SLM12-block | 19.7 | 965M | 20.52 | 50.77 |
> | Fast-SLM 16-block | 20.4 | 533M | 23.34 | 48.40 |
>
> *Note:* Latency is measured on an A100 with an output length of 8k and batch size of 1. CR accuracy is averaged over 8 commonsense reasoning tasks.
>
> ---
>
> **2. Data points in Table 2 and Figure 5**
>
> Thank you for pointing out this potential confusion! As mentioned in Lines 206–207 of our submitted manuscript, our search metric is the model with the lowest PPL under a target latency. Accordingly, the final model we selected corresponds to the one with lowest PPL under approximately 18.3s latency in Figure 5 (which is annotated by a green dot).
>
> The latency values in Figure 5 are obtained by summing the profiled latency of each operator  from the LUT. The 17.71s shown in Table 2 refers to the same model, but with latency measured in an end-to-end manner, which results in a slightly lower absolute value. Although there is a small discrepancy between LUT-based and end-to-end latency measurements, we find that summing the profiled latencies from the LUT preserves the relative latency rankings across different models, particularly when the depth is fixed. This LUT-based approach is also widely adopted in prior NAS works such as FBNet [1]. We will clarify this point in the final version.
>
> ---
>
> **3. The importance of the aspect ratio to SLMs/LLMs**
>
> This is a great question! We have conducted additional analysis on this and plan to emphasize it in the final version. We completely agree that the impact of aspect ratio on both latency and accuracy strongly depends on model size, as elaborated below.
>
> (1) **In terms of latency**, based on our fine-grained profiling on GPUs, the reason why aspect ratios matter for SLMs is that when the model is relatively small, GPU utilization tends to be low with relatively large overheads for kernel launch, making small and large kernels exhibit similar latency. In such scenarios, reducing the number of kernels is key to improving latency, and shallow-wide models can result in fewer but larger kernels. However, for LLMs with much larger kernels, GPU utilization is higher and kernel launch overhead becomes less significant, making aspect ratios less critical.
>
> To demonstrate this, we compare two LLaMA models with 12 and 24 blocks, respectively. We vary their widths to equalize their latency and record the number of parameters, as more parameters may indicate potentially better accuracy. As shown in the table below, with a lower target latency and smaller model sizes, shallow-wide models can yield up to 10x more parameters under the same latency. For larger models, however, the additional parameters enabled by shallower models diminish, potentially resulting in smaller accuracy gains (or even worse accuracy). This indicates that aspect ratio has a greater impact on SLMs.
>
> | Latency (sec.) | Depth=24 - Param (B) | Depth=12 - Param (B) | Increased Param Ratio |
> |:---:|:---:|:---:|:---:|
> | 2.5 | 0.08 | 0.86 | 10.62x |
> | 3.0 | 0.28 | 1.30 | 4.72x |
> | 5.2 | 1.54 | 3.18 | 2.07x |
> | 7.0 | 2.95 | 4.37 | 1.48x |
> | 8.0 | 3.82 | 5.36 | 1.40x |
> | 9.4 | 4.81 | 6.45 | 1.34x |
> | 14.0 | 8.45 | 10.32 | 1.22x |
> | 19.3 | 13.11 | 15.10 | 1.15x |
>
> (2) **In terms of accuracy**, as observed in Figure 2 of our submitted manuscript, as model size increases, the accuracy gaps among different depth–width ratios gradually narrow. This also indicates that aspect ratio matters more for SLMs than for LLMs.
>
> In summary, the above analysis suggests that our techniques and insights regarding depth–width ratios are particularly well-suited for SLM design. We believe this analysis strengthens our paper by identifying when these design factors matter most and when our techniques should be adopted. We will follow your suggestion and add further discussion in the final version.
>
> ---
>
> **4. How to construct the LUT**
>
> We followed common practice [1] to construct the LUT by profiling the latency of each operator, where the operator type and its width (i.e., dimensionality) define an operator, under the target deployment setting. Specifically, we profile Attention, Mamba2, DeltaNet, and FFN with various widths under the target setting and record their latencies in a table. For a model architecture with an arbitrary combination of these operators, we approximate the overall model latency by summing the latency of each operator. We find that this approach preserves the relative latency rankings across different models, particularly when the depth is fixed. We will clarify this in the final version.
>
> [1] “FBNet: Hardware-Aware Efficient ConvNet Design via Differentiable Neural Architecture Search”, B. Wu et al., CVPR 2019.

---

> > ### Comment · Reviewer_X2pZ · 2025-08-05
> > **Thanks for the clarification**
> >
> > Thanks for the clarification.
> >
> > I still hold my assessment for the first point that directly adapting the Llama model's scaling result to hybrid model is not a rigorous approach.
> >
> > For the second point, I suggest to add more clarifications besides Table 2 and Figure 5. This can be fixed in the final version.
> >
> > For the third point, I am more concerned of the accuracy part than the latency part. I would suggest to add a paragraph on the accuracy scaling of SLM vs. LLM for Figure 2.
> >
> > As a summary, I would retain my score between borderline accept to accept.

---

> > > ### Author Response · Authors · 2025-08-05
> > > **Further Response to Reviewer X2pZ**
> > >
> > > Thank you for your response.
> > >
> > > For the first point, we agree that building a separate scaling law for our hybrid model, or jointly searching both operator combinations and depth–width ratios, has greater potential to yield a stronger architecture, and we will apply this to our next architecture.
> > >
> > > For the second point, following your suggestion, we will add the above clarification in the final version.
> > >
> > > For the third point, we will extend Figure 2 to include LLMs in the final version.
> > >
> > > Thank you again for recognizing the potential of our work to “produce significant community impact,” as well as for the constructive comments. We will revise the final version accordingly based on your suggestions.

---

### Official Review · Reviewer_aYwt · 2025-07-03

**Clarity:** 2
**Significance:** 3
**Originality:** 3
**Rating:** 4
**Confidence:** 3

**Summary:**

This paper addresses the gap between parameter efficiency and real-device latency in small language models (SLMs), which is crucial for real-world, latency-sensitive applications. The authors systematically analyze the core determinants of SLM latency, depth-width ratios and choice of operators, and introduce design and training principles for constructing latency-optimal SLMs.

**Questions:**

1. To what extent do your depth-width and operator selection findings transfer to other hardware platforms, such as edge devices, mobile GPUs.

**Ethical Concerns:**

["NO or VERY MINOR ethics concerns only"]

**Final Justification:**

My concerns have been addressed. I will maintain my current score.

**Limitations:**

Yes

**Quality:**

2

**Strengths And Weaknesses:**

Strengths:

1. The paper provides an in-depth examination of the relationship between model architecture (especially depth and width) and real-device latency, an aspect that is underexplored in previous studies.

2. The paper presents an evolutionary search pipeline  to automatically identify complementary combinations of efficient attention operators.

3. The paper is well structured and easy to follow.

Weakness:

1. The developed law is validated on Llama models and a single hardware setup. It is unclear whether the fitted exponents generalise to other architectures such as  MoE architectures or to CPUs/edge NPUs.

2. The latency analysis and all benchmarking are conducted on NVIDIA A100 GPUs, While this is sensible for reproducibility, the broader generalizability of the architectural/principled findings to edge-devices is also important to included considering this paper focus on SLMs.

---

> ### Author Rebuttal · Authors · 2025-07-31
>
> Thank you for your constructive comments! We have addressed all of your questions below:
>
> **1. The generalizability of the architectural findings on other hardware platforms**
>
> Thank you for the suggestions! Following your suggestion, in addition to the data-center GPU A100 reported in our submitted manuscript, we have also verified our findings on (1) a consumer-level GPU, the RTX 3090, and (2) a workstation GPU, the RTX A5000. We will include results on other edge GPUs given more time, after obtaining access to them post-rebuttal.
>
> **a.** Depth-width trade-offs
>
> To demonstrate the importance of depth–width trade-offs and the advantages of shallow-wide models, we fix the model size of a LLaMA model to 500M / 3B and vary its depth and width. We then report the latency on the RTX 3090 and RTX A5000 (batch size = 1, input length = 128, output length = 8000).
>
> As shown in the table below, shallow-wide models consistently exhibit lower latency across both platforms and model sizes, aligning with the observations in Section 2.1 of our submitted manuscript.
>
> | Device | Model Size | 6L | 12L | 18L | 24L | 30L |
> |:---:|:---:|:---:|:---:|:---:|:---:|:---:|
> | RTX 3090 | 500M | 12.99 | 17.88 | 24.52 | 26.44 | 35.58 |
> |  | 3B | 62.10 | 69.34 | 85.87 | 96.05 | 102.31 |
> | RTX A5000 | 500M | 17.48 | 22.03 | 31.17 | 30.78 | 40.71 |
> |  | 3B | 83.73 | 91.02 | 108.28 | 118.80 | 126.03 |
>
>
> **b.** Efficiency of different operators
>
> We benchmark the efficiency of different operators on the RTX 3090 and RTX A5000 (batch sizes = 1 and 16, input length = 128, output length = 8000), following the setting in Figure 4 and Section 2.2 of our submitted manuscript. All models are 500M.
>
> As shown in the table below, Mamba2 and DeltaNet generally stand out as the most efficient operators across settings and platforms, echoing the observations in Section 2.2 of our submitted manuscript. Considering the necessity of attention for recall-intensive tasks, we adopt Attention, Mamba2, and DeltaNet as candidates in our architecture search in Section 2.2.
>
> | Device | Attn | Mamba | Mamba2 | DeltaNet | Gated DeltaNet | GLA |
> |:---:|:---:|:---:|:---:|:---:|:---:|:---:|
> | RTX 3090 (bs=1) | 18.52 | 21.17 | 14.61 | 16.03 | 17.09 | 22.23 |
> | RTX 3090 (bs=16) | 33.38 | 27.54 | 17.06 | 21.98 | 24.21 | 30.21 |
> | RTX A5000 (bs=1) | 22.02 | 25.74 | 18.96 | 20.31 | 20.98 | 26.35 |
> | RTX A5000 (bs=16) | 42.28 | 30.68 | 22.78 | 26.61 | 28.17 | 34.94 |
>
>
> **c.** Benchmark the final model
>
> We further provide benchmarks of the final delivered models on the RTX 3090 and RTX A5000 (batch sizes = 1 and 16, input length = 128, output length = 8000). As shown in the table below, Fast-SLM-2.7B continues to stand out with the lowest latency compared to other models of similar size. Specifically, Fast-SLM-960M and Fast-SLM-2.7B achieve 3.3x and 1.9x speed-ups over Qwen3-0.6B and Qwen2.5-3B, respectively, on the RTX 3090 with batch size = 16.
>
> | Model | Param. | RTX 3090 (bs=1) | RTX 3090 (bs=16) | RTX A5000 (bs=1) | RTX A5000 (bs=16) |
> |:---:|:---:|:---:|:---:|:---:|:---:|
> | SmolLM2-360M | 360M | 32.71 | 62.70 | 34.54 | 76.61 |
> | Qwen2.5-0.5B | 0.5B | 26.68 | 52.89 | 29.86 | 59.46 |
> | Qwen3-0.6B | 0.6B | 38.15 | 131.27 | 42.84 | 164.95 |
> | AMD-OLMo-1B | 1B | 40.69 | 133.45 | 53.94 | 173.07 |
> | Llama-3.2-1B | 1.2B | 32.29 | 62.85 | 43.02 | 77.95 |
> | Fast-SLM-960M (Ours) | 0.96B | 26.61 | 40.29 | 35.39 | 49.56 |
> | Qwen2.5-1.5B | 1.5B | 51.69 | 103.63 | 63.50 | 120.49 |
> | SmolLM2-1.7B | 1.7B | 58.37 | 118.48 | 78.22 | 164.68 |
> | Qwen3-1.7B | 1.7B | 59.82 | 171.93 | 76.81 | 214.85 |
> | IBM-Granite-3.0-2b | 2b | 80.52 | 155.49 | 105.45 | 192.52 |
> | Llama-3.2-3B | 3B | 88.20 | 175.45 | 111.39 | 218.92 |
> | Qwen2.5-3B | 3B | 87.63 | 167.73 | 116.36 | 190.14 |
> | H2O-danube3-4b-base | 4B | 102.38 | 174.21 | 133.58 | 231.66 |
> | Fast-SLM-2.7B (Ours) | 2.7B | 71.93 | 88.12 | 88.81 | 118.50 |
>
> In summary, the key conclusion is that all findings remain valid across different hardware platforms. In general, the low GPU utilization during language model decoding highlights the need for fewer kernel launches, and the quadratic cost of attention motivates the adoption of more efficient operators with linear complexity. These two factors are broadly applicable across GPU platforms, reinforcing the importance of tuning depth–width ratios (as shallow-wide models result in fewer but larger kernels) and leveraging efficient/hybrid operators.
>
> ---
>
> **2. Clarifications on the depth-width scaling law**
>
> We first respectfully clarify that the depth–width scaling law is hardware-agnostic, meaning the fitted exponents are determined solely by the model. Specifically, as shown in Equation 1 of our submitted manuscript, the scaling law predicts the model loss based on depth, width, and data. In practice, given a target latency budget and deployment setting, the optimal depth–width ratio can be identified by profiling a range of configurations and selecting the one that satisfies the latency constraint while minimizing the loss predicted by the scaling law. For models other than Llama, given more time after the rebuttal period, we can follow your suggestion to train a series of models such as MoE and fit the scaling law to observe how the exponents vary.
>
> In addition, we would like to mention that the scaling law derived from Llama can serve as a useful guide for selecting the depth–width ratio of other models. For example, the scaling law on Llama suggests that our Fast-SLM-960M should use 12 blocks, where one block includes one (linear) attention and one FFN. We benchmark Fast-SLM-960M-12b (the adopted configuration) against its 8-block and 16-block variants under an iso-latency setting, i.e., all models are scaled to a comparable latency by adjusting the width.
>
> As shown in the table below, Fast-SLM 12-block achieves the best PPL and accuracy across all three settings, suggesting that the depth–width ratio predicted by the LLaMA scaling law provides a strong starting point. Note that our goal with the scaling law is not to predict a single best setting, but rather to (1) emphasize the general trend of this efficiency-critical design factor, and (2) quickly obtain a near-optimal depth–width ratio under a target latency as a good starting point.
>
> | Model | Latency (sec.) | Param | Wiki PPL | CR Acc (%) |
> |:---:|:---:|:---:|:---:|:---:|
> | Fast-SLM 8-block | 19.5 | 1211M | 20.59 | 50.21 |
> | Fast-SLM12-block | 19.7 | 965M | 20.52 | 50.77 |
> | Fast-SLM 16-block | 20.4 | 533M | 23.34 | 48.40 |
>
> *Note*: Latency is measured on an A100 with an output length of 8k and batch size of 1. CR accuracy is averaged over 8 commonsense reasoning tasks.

---

> > ### Comment · Reviewer_aYwt · 2025-08-07
> >
> > Thank you for the detailed response. My concerns have been addressed. I will maintain my current score.

---

> > > ### Author Response · Authors · 2025-08-07
> > > **Further Response**
> > >
> > > Thank you for your thoughtful feedback and for taking the time to review our response. We’re glad to hear that your concerns have been addressed, and we appreciate your engagement throughout the discussion.

---

### Official Review · Reviewer_eYGu · 2025-07-03

**Clarity:** 2
**Significance:** 2
**Originality:** 2
**Rating:** 5
**Confidence:** 3

**Summary:**

This paper explores what kind of model structure can reduce efficiency while maintaining performance. This paper focuses on depth-width ratio and operator selection. It enhances the scaling law to obtain the best depth-width ratio and proposes an evolutionary search framework to determine the combination of operators. The proposed Fast-SLM achieves better accuracy-latency performance than Llama3.2-3B.

**Questions:**

See "Weaknesses"
1. Fast-SLM failed to reach SOTA
2. Ablation experiments of various technologies of Fast-SLM itself
3. Experiments and analysis of operator combinations.
4. Reasons for choosing these two key factors

**Ethical Concerns:**

["NO or VERY MINOR ethics concerns only"]

**Final Justification:**

Many of my concerns were addressed by the authors, and I encourage them to include new results and interpretations in the final paper.

Thank you for your additional analysis, we would like to raise our rating to 5

**Limitations:**

yes

**Quality:**

3

**Strengths And Weaknesses:**

**Strengths**

This paper explores hybrid SLM models with optimal accuracy-latency through depth-width ratio and operators. The paper also proposes architectural improvements and training techniques, and conducts experiments to analyze their role in SLM training in turn.
This paper proposes a Fast-SLM that combines multiple techniques and achieves better performance than Llama3.2-3B.
In addition, this paper's analysis of the combination of different promising operators may help the community in subsequent model training work.

**Weaknesses**

1. Fast-SLM failed to reach SOTA in 7 out of 9 benchmarks, and even underperformed the 1.5B model in some benchmarks. Although this may be reasonable, as its focus is more on the accuracy-latency trade-off, you still need to explain why this happens.
2. You may need to perform ablation experiments on Fast-SLM itself to better show the impact of each technology on the model, rather than just performing experiments when introducing the technology.
3. More experiments and analysis on operator combinations are needed. As can be seen from Table 1, DeltaNet and Gated DeltaNet have achieved small improvements after adding Mamba2, which makes people wonder whether some of the conclusions in "Observations and Analysis" are statistically significant.
4. What is the reason for the author to choose these two key factors? The latency of real devices is mainly determined by two key factors: the depth and width of the model, and the choice of operators. These two factors are intuitively very reasonable, but I still worry about whether there are omissions. Is this theory proven by previous work, or is it determined by the author based on experience?

---

> ### Author Rebuttal · Authors · 2025-07-31
>
> Thank you for recognizing the insights our work offers to the community, as well as for your constructive comments! We have addressed all of your questions below:
>
> **1. Fast-SLM failed to reach SOTA**
>
> We respectfully clarify that accuracies on certain tasks are heavily influenced by the pretraining data. For example, the Qwen series achieves notably high scores on factual knowledge tasks like MMLU due to their proprietary pretraining corpora. Therefore, in the absence of access to the pretraining data of these models, we believe the average accuracy across diverse tasks offers a more meaningful comparison than single-task performance.
>
> Additionally, since the paper submission, we have continued pretraining our models on an additional 2T tokens (totaling 4T tokens) and benchmarked them against baselines across 14 tasks, including Commonsense Reasoning (MMLU, Lambda, PIQA, ARCC, ARCE, HellaSwag, Winogrande, OBQA), Math (GSM8k, MathQA), and Coding (HumanEval, HumanEval-Plus, MBPP, MBPP-Plus).
>
> As shown in the table below, our Fast-SLM-2.7B achieves SOTA average accuracy among all models. Specifically, Fast-SLM-960M surpasses Qwen3-0.6B by 3.77% in average accuracy while being 1.75× faster. Fast-SLM-2.7B outperforms LLaMA3.2-3B and Qwen2.5-3B by 9.24% and 0.90% in average accuracy, respectively, with 1.46x and 1.67x speed-ups. We will include these updated results in the final version and open-source our model.
>
>
> | Model | Param. | Dec. Latency (s) | 8k Cache Size (MB) | CR (%) | Math (%) | Coding (%) | Avg (%) |
> |:---:|:---:|:---:|:---:|:---:|:---:|:---:|:---:|
> | AMD-OLMo-1B | 1B | 28.39 | 1049 | 49.21 | 12.66 | 7.35 | 32.03 |
> | Llama-3.2-1B | 1.2B | 28.37 | 262 | 49.73 | 17.24 | 24.15 | 37.78 |
> | Qwen2.5-0.5B | 0.5B | 28.19 | 98 | 48.13 | 32.66 | 32.06 | 41.33 |
> | Qwen3-0.6B | 0.6B | 33.91 | 918 | 49.92 | **36.88** | 24.32 | 40.74 |
> | Fast-SLM-960M (Ours) | 0.96B | **19.36** | **36** | **52.54** | 24.57 | **38.44** | **44.51** |
> | h2o-danube3-4b-base | 4B | 61.94 | 737 | 60.85 | 34.73 | 11.65 | 43.06 |
> | SmolLM2-1.7B | 1.7B | 41.55 | 1573 | 58.64 | 32.88 | 21.11 | 44.24 |
> | Llama-3.2-3B | 3B | 60.38 | 918 | 59.37 | 30.07 | 34.80 | 48.16 |
> | Qwen2.5-1.5B | 1.5B | 44.06 | 229 | 57.53 | 48.14 | 42.95 | 52.02 |
> | Qwen3-1.7B | 1.7B | 46.39 | 918 | 58.62 | 53.71 | 43.76 | 53.67 |
> | Qwen2.5-3B | 3B | 69.19 | 295 | 60.69 | **53.83** | 49.46 | 56.50 |
> | Fast-SLM-2.7B (Ours) | 2.7B | **41.31** | **80** | **61.62** | 48.88 | **53.22** | **57.40** |
>
> ---
>
> **2. Ablation experiments on Fast-SLM’s components**
>
> Thank you for the suggestion! We believe this analysis can strengthen our paper. Following your suggestion, we conducted the following ablation study: Starting from a 500M LLaMA model with 24 attention blocks, we progressively reach our final Fast-SLM through the following steps: (1) tune the depth/width ratio, (2) enhance it into Fast-SLM’s hybrid architecture, (3) add meta tokens, and (4) add weight normalization.
>
> As shown in the table below, we observe that (1) tuning the depth/width ratio (into a shallow-wide model) reduces latency even though the parameter count is almost doubled, leading to better accuracy; (2) enhancing it into Fast-SLM’s hybrid architecture boosts task accuracy while significantly improving efficiency, e.g., a 3.2x latency reduction for (bs=32, output len=8k) generation; (3) adding meta tokens and weight normalization further improves task performance, yielding a +2.27% accuracy gain in total.
>
> We will include this analysis in the final version of our paper.
>
> | Technique | Model Arch | Param | Latency (bs=1, output_len=8k) | Latency (bs=32, output_len=8k) | Wiki PPL | Acc (%) |
> |:---:|:---:|:---:|:---:|:---:|:---:|:---:|
> | [*Starting point*] | Llama3 24-Block | 500M | 30.41 | 110.77 | 23.74 | 48.18 |
> | + Tuned depth-width ratio | Llama3 12-Block | 970M | 24.75 | 101.11 | 20.96 | 50.11 |
> | + Hybrid operators | Fast-SLM | 965M | 19.46 | 31.87 | 20.52 | 50.77 |
> | + Meta tokens | Fast-SLM | 965M | 19.54 | 32.33 | 20.38 | 50.88 |
> | + Weight normalization | Fast-SLM | 965M | 19.54 | 32.33 | 19.42 | 53.04 |
>
> *Note*: Acc is averaged over 8 commonsense reasoning tasks.
>
> ---
>
> **3. More analysis of operator combinations**
>
> As a recap of the context, for the 500M architecture ablation in Table 1 of our submitted manuscript, we find that adding Mamba2 with other operators can result in a 0.43~1.12 PPL reduction. For example, combining DeltaNet and Mamba2 leads to a 0.5/0.99 PPL reduction and a +0.2%/+0.3% accuracy improvement over pure DeltaNet and pure Mamba2, respectively. We humbly clarify that this improvement is non-trivial, as the reported gains of emerging operators over prior baselines are often of a similar scale. For example, DeltaNet [1] improves over its Mamba baseline by -0.15 PPL / +0.3% accuracy (see their Table 1 at 340M scale).
>
> In addition, we have extensively experimented with different operator combinations, and in general, these hybrid architectures do not consistently outperform the pure models, often resulting in comparable PPL. The benefit of adding Mamba2 is a particularly interesting observation that we have consistently seen, and we highlight this aspect to provide insights to the community.
>
> It is also worth noting that the goal of this combination study is to identify promising operators that are efficient and complementary to one another to serve as a search space for evolutionary search, rather than to find the single best hybrid architecture. Following your suggestion, we will include the comprehensive exploration of different operator combinations we did in the appendix of the final version.
>
>
> [1] “Parallelizing Linear Transformers with the Delta Rule over Sequence Length”, S. Yang et al., NeurIPS 2024.
>
> ---
>
> **4. Reasons for choosing these two key factors**
>
> Thank you for the question! We also plan to emphasize this in the final version.
>
> In general, there are two critical efficiency metrics for SLMs (corresponding to two use cases): (1) inference latency under batch size 1, and (2) generation throughput with the maximum batch size without out-of-memory.
>
> For case (1), GPU utilization is often low with large overheads for kernel launch, making small and large kernels exhibit similar latency. In such scenarios, reducing the number of kernels is key to improving latency. Therefore, tuning the depth–width ratio, where shallow-wide models result in fewer but larger kernels, is crucial for optimizing batch size = 1 latency.
>
> For case (2), GPUs are better saturated and become compute-bound, so adopting compute-efficient operators is critical. Building hybrid operators with efficient linear attention is an effective approach. (These hybrid operators can also help in case (1) by reducing latency per kernel.)
>
> By optimizing these two macro-level design factors, our Fast-SLM-2.7B achieves efficiency across a wide range of deployment scenarios. For example, it is 1.67x / 3.34x faster than Qwen2.5-3B in case (1) and case (2), respectively. There may be other finer-grained or minor design factors, but we believe the above insights can also help guide those decisions. We will highlight this discussion in the final version.

---

> > ### Comment · Reviewer_eYGu · 2025-08-07
> >
> > Many of my concerns were addressed by the authors, and I encourage them to include new results and interpretations in the final paper.
> >
> > Thank you for your additional analysis, we would like to raise our rating to 5

---

> > > ### Author Response · Authors · 2025-08-07
> > > **Further Response**
> > >
> > > Thank you for your thoughtful feedback and for acknowledging our additional analysis! We will incorporate the new results and interpretations into the final version.

---

> ### Comment · Area_Chair_54jA · 2025-08-05
> **Dear reviewer, please read the rebuttal and participate in discussions with authors. Thank you**
>
> Dear reviewer, please read the rebuttal and participate in discussions with authors. Thank you

---

> > ### Comment · Reviewer_eYGu · 2025-08-07
> >
> > already done

---

### Note · Authors · 2025-08-14

We first thank the reviewers for recognizing the insights of our work and its potential impact on the community. For example, Reviewer X2pZ described our work as “*a decent attempt towards latency-optimal SLM*” and noted its potential to “*produce significant community impact*,” while Reviewer ictF highlighted the “*useful insights*,” “*strong results*,” and “*practical significance*” of our work.

We also thank the AC and the reviewers for their constructive comments. We have addressed all questions from all reviewers and will incorporate the changes in our final version. Below, we summarize the highlights from our responses to the reviewers’ common questions:

---

**1. Results on More Tasks, Benchmark with SOTA, and Ablation Study of Each Component**

As detailed in our responses to Reviewer eYGu (Q1), we benchmarked our Fast-SLM models against baselines across 14 tasks, including commonsense reasoning, math, and coding. Our Fast-SLM-2.7B achieves SOTA average accuracy across all tasks, which outperforms LLaMA3.2-3B and Qwen2.5-3B by 9.24% and 0.90% in average accuracy, respectively, with 1.46x and 1.67x speed-ups.

In addition, we have benchmarked on long-context benchmarks and performed a detailed ablation study of each component of Fast-SLM. Please refer to our responses to Reviewers eYGu (Q2) and ictF (Q4).

---

**2. Generality of the Findings on Other Hardware Platforms**

As detailed in our responses to Reviewer aYwt (Q1), we have verified our findings regarding depth-width ratios and operator combinations on an RTX 3090 GPU and an RTX A5000 GPU, in addition to the A100 GPU used in the submitted manuscript. The key conclusions are that (1) all findings remain valid across different hardware platforms, and (2) Fast-SLM consistently outperforms baselines in both speed and accuracy, e.g., Fast-SLM-960M and Fast-SLM-2.7B achieve 3.3× and 1.9× speed-ups over Qwen3-0.6B and Qwen2.5-3B, respectively, on the RTX 3090 with a batch size of 16.

We have also conducted an ablation on the deployment setting of Fast-SLM in our response to Reviewer ictF (Q5) and shown that our built deployment flow with CUDA Graph, which supports arbitrary operators across all GPU platforms, is another infrastructure-level contribution to the community.

---

For all other questions, please refer to our responses to each reviewer. We again thank all reviewers and commit to open-sourcing our final models, along with all search and training recipes, upon acceptance.

---

### Decision · Program_Chairs · 2025-09-17

**Decision:**

Accept (poster)

**Comment:**

Based on the comprehensive review process and author responses, this paper presents a valuable contribution to efficient small language model design by systematically addressing the gap between parameter efficiency and real-device latency. The authors convincingly demonstrated the importance of optimizing depth-width ratios and hybrid operator selection through novel scaling laws and evolutionary search, supported by extensive experiments across multiple hardware platforms. While initial concerns regarding generalizability, task diversity, and novelty were raised, the authors addressed these through additional benchmarks on math, coding, and long-context tasks, ablation studies, and clarification of deployment dependencies. The resulting Fast-SLM models achieve state-of-the-art accuracy-latency trade-offs, with significant speed-ups and cache reductions compared to strong baselines. The commitment to open-sourcing models and methodologies further enhances the work's practical impact. The reviewer consensus supports acceptance given the paper's technical rigor, empirical validation, and potential to influence efficient language model deployment.